# Pancreatic cancer-associated fibroblasts modulate macrophage differentiation via sialic acid-Siglec interactions
Kelly Boelaars [1,2,3], Ernesto Rodriguez [1,2,3], Zowi R. Huinen[1,2,3], Chang Liu [1,2,3,4], Di Wang[5], Babet O. Springer[1,2,3], Katarzyna Olesek[1,2,3], Laura Goossens-Kruijssen[1,2,3], Thomas van Ee[1,2,3], Dimitri Lindijer[1,2,3], Willemijn Tak[1,2,3], Aram de Haas[1,2,3], Laetitia Wehry[1,2,3], Joline P. Nugteren-Boogaard[1,3], Aleksandra Mikula[1,2,3], Charlotte M. de Winde [1,2,3], Reina E. Mebius[1,2,3], David A. Tuveson[6], Elisa Giovannetti [2,7,8], Maarten F. Bijlsma[2,9,10], Manfred Wuhrer [5], Sandra J. van Vliet[1,2,3] & Yvette van Kooyk [1,2,3] ✉

Despite recent advances in cancer immunotherapy, pancreatic ductal adenocarcinoma (PDAC) remains unresponsive due to an immunosuppressive tumor microenvironment, which is characterized by the abundance of cancer-associated fibroblasts (CAFs). Once identified, CAF-mediated immune inhibitory mechanisms could be exploited for cancer immunotherapy. Siglec receptors are increasingly recognized as immune checkpoints, and their ligands, sialic acids, are known to be overexpressed by cancer cells. Here, we unveil a previously unrecognized role of sialic acid-containing glycans on PDAC CAFs as crucial modulators of myeloid cells. Using multiplex immunohistochemistry and transcriptomics, we show that PDAC stroma is enriched in sialic acid-containing glycans compared to tumor cells and normal fibroblasts, and characterized by ST3GAL4 expression. We demonstrate that sialic acids on CAF cell lines serve as ligands for Siglec-7, -9, -10 and -15, distinct from the ligands on tumor cells, and that these receptors are found on myeloid cells in the stroma of PDAC biopsies. Furthermore, we show that CAFs drive the differentiation of monocytes to immunosuppressive tumor-associated macrophages in vitro, and that CAF sialylation plays a dominant role in this process compared to tumor cell sialylation. Collectively, our findings unravel sialic acids as a mechanism of CAF-mediated immunomodulation, which may provide targets for immunotherapy in PDAC.

Pancreatic Ductal Adenocarcinoma (PDAC) is one of the most aggressive types of cancer, with a 5-year survival of around 11%[1]. This poor prognosis is mainly attributed to late diagnosis and limited treatment possibilities. While immunotherapy such as immune checkpoint blockade (ICB) has improved the survival of patients in a range of human cancers, immunotherapy remains unsuccessful in PDAC due to a relatively low mutational burden and various immunosuppressive mechanisms[2]. Understanding the driving factors of immunosuppression is essential to improve treatment possibilities for PDAC.

[1]Amsterdam UMC location Vrije Universiteit Amsterdam, Molecular Cell Biology and Immunology, De Boelelaan, 1117 Amsterdam, Netherlands. [2]Cancer Center Amsterdam, Cancer Biology and Immunology, Amsterdam, The Netherlands. [3]Amsterdam Institute for Infection and Immunity, Cancer Immunology, Amsterdam, The Netherlands. [4]Amsterdam UMC location Vrije Universiteit Amsterdam, Pulmonary Medicine, De Boelelaan, 1117 Amsterdam, the Netherlands. [5]Leiden University Medical Center, Center for Proteomics and Metabolomics, Albinusdreef 2, 2333 ZA Leiden, the Netherlands. [6]Laboratory, Cold Spring Harbor, New York, USA. [7]Amsterdam UMC location Vrije Universiteit Amsterdam, Medical Oncology, De Boelelaan, 1117 Amsterdam, Netherlands. [8]Cancer Pharmacology Lab, AIRC Start-Up Unit, Fondazione Pisana per la Scienza, Pisa, Italy. [9]Amsterdam UMC, location University of Amsterdam, Center for Experimental and Molecular Medicine, Laboratory for Experimental Oncology and Radiobiology, Meibergdreef 9, 1105AZ Amsterdam, The Netherlands. [10]Oncode Institute, Amsterdam, The Netherlands. ✉e-mail: y.vankooyk@amsterdamumc.nl

The PDAC tumor microenvironment (TME) is unique in its abundance of dense fibrotic stroma and suppressive immune cells. The stroma in PDAC can constitute up to 80% of the tumor mass and comprises of extracellular matrix and specialized connective-tissue cells, including cancer-associated fibroblasts (CAFs)[3]. CAFs are highly heterogeneous in their phenotypes, origins and functions, including both tumor-promoting and tumor-inhibiting properties[4-9]. Advances in single-cell technologies have led to the identification of several CAF subsets in PDAC that include inflammatory CAFs (iCAF), myofibroblastic CAFs (myCAF) and antigen-presenting CAFs (apCAF)[10-14]. Immune cells in the TME are mainly of the myeloid lineage, the majority being tumor-associated macrophages (TAMs)[15,16]. Accumulation of TAMs correlates with poor prognosis of PDAC[16-18]. TAMs are chronically polarized by the tumor and show a mixed phenotype of both anti-tumoral and pro-tumoral activation states, marked by the expression of HLA-DR and CD86 or CD163, CD206 and PD-L1, respectively[15,19]. Besides the role of TAMs in tissue remodeling and inflammation, TAMs induce immunosuppression in the TME by recruiting Tregs and inhibiting CD8$^+$ T and NK cell cytotoxicity through secretion of cytokines such as TGF-β and interleukin (IL)-10 [15,20-25].

Accumulating evidence shows that the crosstalk between CAFs and immune cells leads to inhibition of anti-tumor immune responses and can hamper immunotherapy[26]. In breast, prostate, skin and colorectal cancer, CAFs can recruit monocytes through secretion of CCL2 and enhance their polarization to immunosuppressive TAMs[26]. In addition, co-injection of Panc02 PDAC tumor cells with stellate cells increased the accumulation of myeloid cells, including pro-tumoral TAMs in the tumor[27]. Moreover, depletion of fibroblast activating protein (FAP)$^+$ CAFs in murine PDAC improved the efficacy of ICB and reduced tumor growth[28]. On the other hand, depletion of alpha-smooth muscle actin (α-SMA)$^+$ CAFs accelerated tumor growth but when combined with ICB prolonged the survival of mice[6]. These studies highlight a role of CAFs in immunosuppression potentially via myeloid cells. Yet, it is still unclear whether PDAC CAFs can directly induce TAMs with immunosuppressive properties and what are the mechanisms behind this. A better understanding of CAF-mediated myeloid suppression can contribute to developing more effective immunotherapeutic strategies in PDAC.

Changes in metabolic processes such as glycosylation are a hallmark of cancer and have been shown to alter immune responses via binding to lectin receptors[29-31]. One of the most observed glycan alterations in cancer is the overexpression of sialic acids[29,31,32]. The synthesis of sialylated glycans occurs in the Golgi by sialyltransferase enzymes (ST) that utilize the sialic acid donor cytidine monophosphate *N*-acetylneuraminic acid (CMP-Neu5Ac) generated by the Cytidine Monophosphate *N*-Acetylneuraminic Acid Synthetase (CMAS) in the nucleus[33]. Tumor cells overexpress sialic acids to evade immune clearance and create an immunosuppressive environment[29,31,32,34]. Sialic acids are therefore considered a target for immunotherapy[34]. Sialic acids can induce immunosuppression by binding to ITIM-containing Sialic acid-binding immunoglobulin-like lectins (Siglecs) expressed on innate and adaptive immune cells[32,35]. In cancer, Siglec-7, -9, -10 and -15 play a key role in facilitating immune evasion[36-41]. The binding of Siglecs to sialic acid is complex and dependent on multiple factors including, sialic acid linkage, underlying glycan structure and on the protein or lipid it is attached to[41,42].

Increasing evidence from mouse models highlights the immunosuppressive role of tumor sialylation[43-47]. Tumor sialylation suppresses CD8$^+$ T cell and NK cell cytotoxicity and polarizes macrophages to immunosuppressive macrophages, both supporting tumor growth[43-48]. Targeting tumor sialylation, either through sialic acid mimetics or sialidase-coupled antibodies, improves survival of mice and synergizes with ICB, which is dependent on Siglec-E, the murine orthologs of Siglec-7 and Siglec-9[45-48]. Our previous study showed that overexpression of sialic acids in PDAC cells contributes to an immunosuppressive microenvironment by promoting TAM differentiation via the interaction with Siglec-7 and Siglec-9[36]. These studies highlight the potential of targeting tumor sialylation for cancer immunotherapy. However, there is a limited understanding of the

glycosylation and sialic acid expression on PDAC CAFs, and how this can mediate Siglec-dependent immunosuppression.

Here we study sialylation of CAFs in relation to tumor sialylation, and its role in immune modulation of myeloid cells in PDAC. We report increased sialylation of CAFs in PDAC compared to tumor cells and normal fibroblasts. PDAC CAFs induce the differentiation of monocytes to immunosuppressive TAMs, a process in which CAF derived sialic acids plays a significant role. These data, coupled with the finding that the majority ( ~ 90%) of myeloid cells reside in the stroma, highlight that CAF sialylation plays a dominant role in TAM-differentiation when compared to tumor cell sialylation.

## Results
### PDAC stroma expresses sialylated glycan structures
We and others have previously demonstrated that PDAC tumor sialylation drives myeloid suppression through interactions with Siglec-7 and Siglec-9[36,39,40]. Interestingly, DAB staining for sialic acid in PDAC biopsies revealed its presence not only in tumor cells but also in the stromal compartment (Fig. 1a, Supplementary Fig. 1a). The presence of sialic acids was assessed by staining with an inactivated neuraminidase (Lectenz), that has specificity for all α2-3, α2-6 and α2-8 sialic acid linkages (pan-Lectenz). The specificity for sialylated structures was confirmed by pre-treating the tissue with neuraminidase, an enzyme that hydrolyses terminal sialic acids (Supplementary Fig. 1b). To quantify sialic acid levels, PDAC biopsies were co-stained with the tumor marker panCK and pan-Lectenz, allowing for the segmentation of tumor and stromal areas. Interestingly, the median staining intensity of pan-Lectenz was significantly higher in stromal regions than on the panCK$^+$ tumor cells, underscoring the abundance of sialic acids in the cancer stroma (Fig. 1b).

Given the prevalence of CAFs in PDAC stroma, we hypothesized that CAFs express sialic acids which could drive Siglec-dependent immune modulation. To explore this hypothesis, we analyzed several publicly available transcriptomic datasets of PDAC, focusing on genes involved in sialic acid-containing glycan production (Fig. 1c). These genes include those responsible for donor synthesis and transportation (*GNE, NANS, NANP, CMAS, SLC35A1*) and sialyltranferases expressed in the Golgi. Sialyltransferases are categorized based on acceptor sugar, which is a galactose (Gal), N-acetylgalactosamine (GalNAc) or another sialic acid (Sia), and the linkage (α2-3, α2-6, α2-8) in which they add sialic acids (Fig. 1c). Correlating gene signatures of different sialylation pathways with CAFs and tumor cell scores in the TCGA bulk RNA sequencing dataset revealed a positive correlation between the overall sialylation-related gene set and the CAF-gene signature score, while the tumor cell score showed no significant correlation (Fig. 1d). Additionally, pathways involving α2-3 or α2-6 sialylation positively correlated with CAFs, but not with tumor cells, which instead showed a positive correlation only with donor synthesis genes (Fig. 1d). Using a publicly available RNA sequencing dataset of microdissected tumor and stroma[49], we identified individual sialylation genes associated with stromal sialylation. Tumor cells highly expressed donor synthesis genes (*NANS, CMAS*, and the transporter *SLC35A1*), while the stroma showed enriched expression of sialyltransferases involved in α2-3, α2-6 and α2-8 sialylation (*ST3GAL2, ST3GAL5, ST6GAL2, ST6GALNAC5, ST6GALNAC6, ST8SIA1* and *ST8SIA2*) (Fig. 1e). These findings imply that stromal cells can produce sialylated glycans as they express the genes involved in sialylation.

To assess the expression of sialylation-related genes within distinct TME cell subsets, including CAFs, we utilized publicly available scRNA-Seq data from Peng et al.[50]. Cell clusters were identified, and the expression of sialylation genes was analyzed for each cluster (Fig. 1f, Supplementary Fig. 1c, d). Interestingly, we identified the sialyltransferase *ST3GAL4* to be highly expressed in the fibroblast cell cluster compared to other cell clusters (Fig. 1g, Supplementary Fig. 1d). ST3GAL4 is one of the six enzymes that transfers sialic acids in a α2-3 linkage to terminal galactopyranosyl (Gal). ST3GAL4 is involved in generation of the sialyl Lewis X antigen, often upregulated in tumor cells to facilitate invasion, and is described to generate the ligands for Siglec-9[36,51]. Another sialylation enzyme enriched in stromal cells was ST6GALNAC6, but this enzyme was mainly expressed in smooth

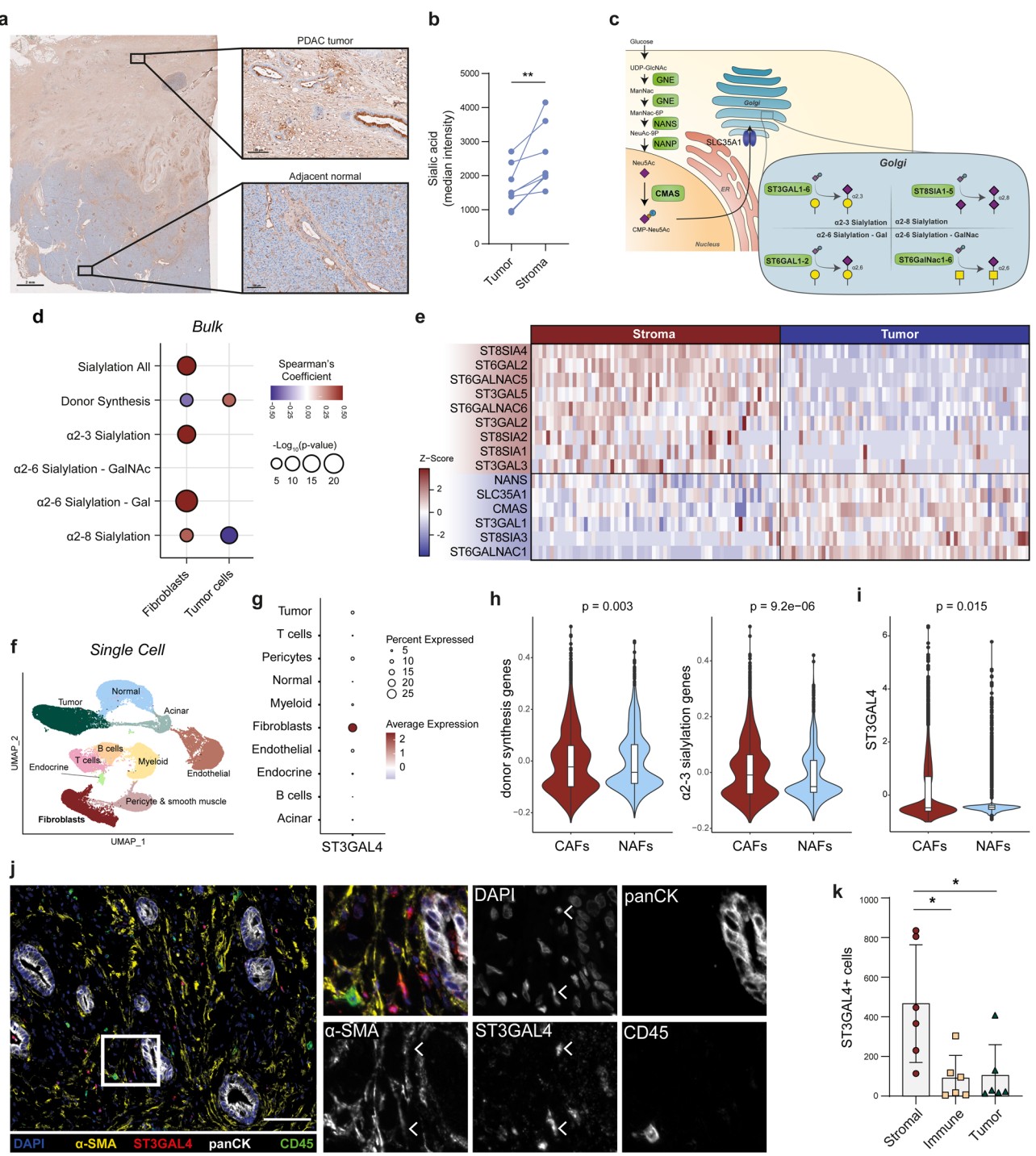

muscle cells or pericytes (Supplementary Fig. 1d, e). Further examination revealed several fibroblast subsets, that we annotated as normal-fibroblast (NAFs), given their presence in normal pancreatic tissue, transitional-CAFs, myofibroblastic CAFs (myCAFs) and inflammatory-CAFs (iCAFs) (Supplementary Fig. 1f, g). Interestingly, both donor synthesis and α2-3 sialylation genes were significantly enriched in CAFs compared to NAFs (Fig. 1h). We did not identify significant differences in α2-3 sialylation genes between transitional CAFs, iCAFs or myCAFs, suggesting that sialylation is an overall signature of all CAF subsets (Supplementary Fig. 1h). *ST3GAL4* was upregulated in CAFs compared to NAFs and was significantly higher expressed in myCAFs compared to iCAFs (Fig. 1i, Supplementary Fig. 1h). To validate ST3GAL4 expression in CAFs, PDAC biopsies were co-stained

for the CAF marker α-SMA and ST3GAL4. Indeed, ST3GAL4 was expressed in several α-SMA⁺ cells in the PDAC tissues (Fig. 1j). Quantification of the ST3GAL4 staining confirmed that ST3GAL4 was predominantly expressed in stromal cells (Fig. 1k). In summary, these data demonstrate that PDAC CAFs express sialic acids and are characterized by the expression of the sialyltransferase ST3GAL4.

### Sialic acids on CAF cell lines serve as ligands for Siglec receptors expressed on myeloid cells in the TME

To further study the sialylation of CAFs, we utilized three fibroblast cell lines (M1 CAFs, T1 CAFs, and PS-1) to model pancreatic CAFs (Supplementary Fig. 2a). The cell lines were characterized by the lab of origin,

**Fig. 1 | Sialic acid-containing glycans are expressed in the tumor stroma, characterized by the expression of ST3GAL4 in CAFs. a** Sialic acid expression in PDAC patient biopsy, as assessed by staining with pan-Lectenz (*n* = 8, see Fig. S1a), which binds all sialic acid linkages. Scale bar on left image equals 2 mm, scale bar on zoomed in images on the right equals 100 μm. **b** Median intensity of pan-Lectenz staining for sialic acids in PDAC patient biopsies, comparing tumor area (panCK + ) with stroma (panCK-) (*n* = 8). Statistical analysis with paired student-t test. **c** Schematic representation of sialylation-related genes involved in sialic acid metabolism and the sialyltransferases involved in the transfer of sialic acids to glycans. **d** Correlation between different sialic acid-related gene signatures and cell signatures of fibroblast or tumor cells in PDAC patients, analyzed using a bulk-RNA sequencing dataset from TCGA. **e** Heatmap of significant differentially expressed sialylation genes comparing microdissected tumor versus stroma in a RNA-seq dataset from Maurer et al.[49]. **f** UMAP of scRNA-seq data of PDAC patients from Peng et al.[50] illustrating the different cell populations identified in this dataset. **g** Expression of the sialyltransferase enzyme *ST3GAL4* in the scRNA-seq dataset

from (**e**) across cell type clusters. **h** Sialylation gene scores in CAFs versus NAFs from scRNA-seq data from (**e**). Statistical analysis using Wilcoxon test. Data presented as boxplot indicate the median, 25th and 75th percentiles (hinges) and whiskers represent 1.5 times the interquantile range. **i** ST3GAL4 expression in CAFs versus NAFs from scRNA-seq data from (**e**). Statistical analysis using Wilcoxon test. Data presented as boxplot indicate the median, 25th and 75th percentiles (hinges) and whiskers represent 1,5 times the interquantile range. **j** Multiplex immunohistochemistry on PDAC biopsies stained for nuclei (DAPI), immune cells (CD45), tumor cells (panCK), cancer-associated fibroblasts (α-SMA) and the sialyltransferase ST3GAL4. White arrows indicate expression of ST3GAL4 in α-SMA⁺ cells (representative image of *n* = 6). Scale bar equals 100 μm. **k** Quantification of ST3GAL4⁺ cells within stroma, immune cells and tumor cells. Tumor and stromal area's divided based on panCK signal. Tumor cells were identified with panCK, immune cells in the stroma identified based on CD45 expression, stromal cell identified as panCK- and CD45- within the stromal region. Statistical analysis with one-way repeated measures ANOVA and dunnett's multiple comparison test.

showing lack of cancer driver mutations in M1 and T1 CAFs[14,52]. Although pancreatic stellate cells are typically quiescent cells in homeostasis, they are known to acquire an activated myofibroblast phenotype when cultured in monolayer or when activated in the TME, and therefore can be used as a model for CAFs[4,14]. Indeed, the PS-1 cell line expressed the myofibroblast marker α-SMA, lost the quiescent marker GFAP, and displayed an activated phenotype, and will therefore be referred to as PS-1 CAF (Supplementary Fig. 2b–d). The mesenchymal origin of these CAF cell lines was confirmed with vimentin staining, and all cell lines expressed the CAF markers α-SMA, FAP and CD90 (Fig. 2a, Supplementary Fig. 2b, e, f)[13]. Nevertheless, the three CAF cell lines showed distinct properties and activation states. Firstly, The M1 CAFs and PS-1 CAFs displayed an elongated, aligned phenotype compared to T1 CAFs (Fig. 2a). Additionally, the PS-1 CAFs were significantly more activated compared to the M1 and T1 CAFs, and several fibroblast markers were differentially expressed in the cell lines (Supplementary Fig. 2c, d, f). Given the differential phenotype and activation state of the cells, we next aimed to evaluate whether the cells resemble iCAF or myCAFs. Therefore, we analyzed a set of fibroblast markers and compared their expression to the expression in CAF subsets in scRNA-seq data, as well as their expression on the CAF cell lines after treatment with IL-1β or TGF-β, inducing an iCAF or myCAF phenotype respectively. While PDGFRα has been described as an iCAF marker[13], our analysis associated it with both iCAFs and normal fibroblasts, suggesting that this marker may not be specific to iCAFs. In line with this, treatment of the cell lines with IL-1β, a cytokine described to induce an iCAF phenotype[53], slightly decreased PDGFRα expression (Supplementary Fig. 2f). PDGFRα was expressed in M1 and T1 CAFs, but not in PS-1 CAFs. Additionally, the cytokine secretion profiles of the CAFs were analyzed, as iCAFs are characterized by secretion of IL-6, CCL2 and CXCL12[13,53] (Supplementary Fig. 2g). All CAF cells secreted high levels of IL-6, IL-8, CCL2 (>5000 pg/mL), suggesting an iCAF-like phenotype in all cell lines (Supplementary Fig. 2g). Expression of IL-6 and IL-8 were over 2-fold higher in the PS-1 CAFs compared to the M1 and T1 CAFs, suggesting a more iCAF-like phenotype in the PS-1. In contrast, all cell lines expressed the myCAF marker α-SMA, with the PS-1 being most myCAF-like given its high myofibroblastic, activated phenotype in the gel contraction assay (Supplementary Fig. 2b, d). Furthermore, all cell lines expressed PD-L1 and did not represent apCAFs, as evidenced by the lack of HLA-DR (Supplementary Fig. 2h). Together, the phenotyping of the cell lines indicates that the fibroblast cell lines are suitable for in vitro modeling of CAFs in PDAC, and show characteristics of both iCAFs and myCAFs.

We next examined the sialic acid expression in fibroblast and tumor cell lines using various biotinylated probes. Staining with pan-Lectenz or α2-3-specific-Lectenz revealed that all fibroblast cell lines exhibited equal or higher sialic acid levels than BxPC3 and MiaPaca-2 tumor cell lines (Fig. 2b, Supplementary Fig. 2i). Additionally, staining with plant lectin derived from *Maackia amurensis* (MAAII) or *Sambucus nigra* (SNA), with specificity for

α2-3 and α2-6 linked sialic acids respectively, demonstrated sialic acid expression in the cell lines, with higher binding to α2-3 linked sialic acids compared to α2-6 linked sialic acids in the M1 and T1 CAFs (Fig. 2c, Supplementary Fig. 2i).

Sialic acids have been widely described as immunosuppressive carbohydrates, acting as ligands for inhibitory Siglec receptors on immune cells. To investigate which Siglec receptors could potentially bind CAF sialic acids, we stained the CAF cell lines with different Siglec-Fc chimeras. We have demonstrated before that tumor sialic acids interact with Siglec-7 and Siglec-9[36]. Here we show that among the tested Siglecs, Siglec-7, -9, -10 and -15 recognized CAF sialic acids (Fig. 2d). The binding of these Siglec-Fcs was sialic acid specific, as treatment with the enzyme neuraminidase abolished Siglec-Fc binding (Fig. 2d). Other Siglec receptors, including Siglec-3, -5, -6, -8, -11 and -14, did not bind CAF sialic acids (Supplementary Fig. 2j). Taken together, pancreatic CAFs express sialylated glycans in a variety of linkages, with α2-3 linked being the most predominant, and these sialic acids serve as ligands for Siglec-7, -9, -10 and -15.

Siglecs are present on various immune cells, such as Siglec-7 and -9 on NK cells[54,55], Siglec-9 on a subset of T cells[56,57], and Siglec-7, -9, -10 and -15 on tumor-associated macrophages (TAMs)[36–38]. Given that myeloid cells can express Siglec-7, -9, -10 and -15[32,36], and are a dominating immune subset in the TME of PDAC[58], we investigated the presence of these Siglecs on CD14⁺ myeloid cells in the TME. Multiplex immunohistochemistry revealed the existence of CD14⁺ myeloid cells expressing Siglec-7, -9, -10 or -15 in the PDAC TME (Fig. 2e). Interestingly, most CD14⁺ cells were located in the stroma, suggesting potential interactions between stromal cells and myeloid cells (Fig. 2f). Across the examined PDAC biopsies, Siglec-9 emerged as the most abundantly expressed Siglec within the CD14⁺ cells, followed by Siglec-10, Siglec-7 and Siglec-15 (Fig. 2g). These data indicate the expression of Siglec-7, -9, -10 and -15 on myeloid cells in PDAC, potentially facilitating interactions with sialic acids on CAFs.

To evaluate whether these receptors show differential expression among distinct monocyte-derived cells, including different macrophage activation and polarization states, we generated monocyte-derived macrophages (moMACs) and monocyte-derived dendritic cells (moDCs) in vitro. MoMACs were generated by differentiating monocytes with M-CSF, creating non-polarized moMACs (M0), and were polarized to M1-moMACs using LPS and IFNγ or M2-moMACs using IL-4 and IL-6. MoDCs were generated by addition of GM-CSF and IL-4. Among these cells, Siglec-7, -9 and to some extend Siglec-10 were expressed on undifferentiated monocytes, moMACs and moDCs. Interestingly, Siglec-15 was expressed on M0- and M2-moMACs, but not on undifferentiated monocytes, M1-moMACs or moDCs (Supplementary Fig. 3a).

## Detailed characterization of glycan profiles in PDAC CAF cell lines

Because of a limited understanding of specific glycans present on CAFs, we conducted a comprehensive glycan analysis using mass spectrometry,

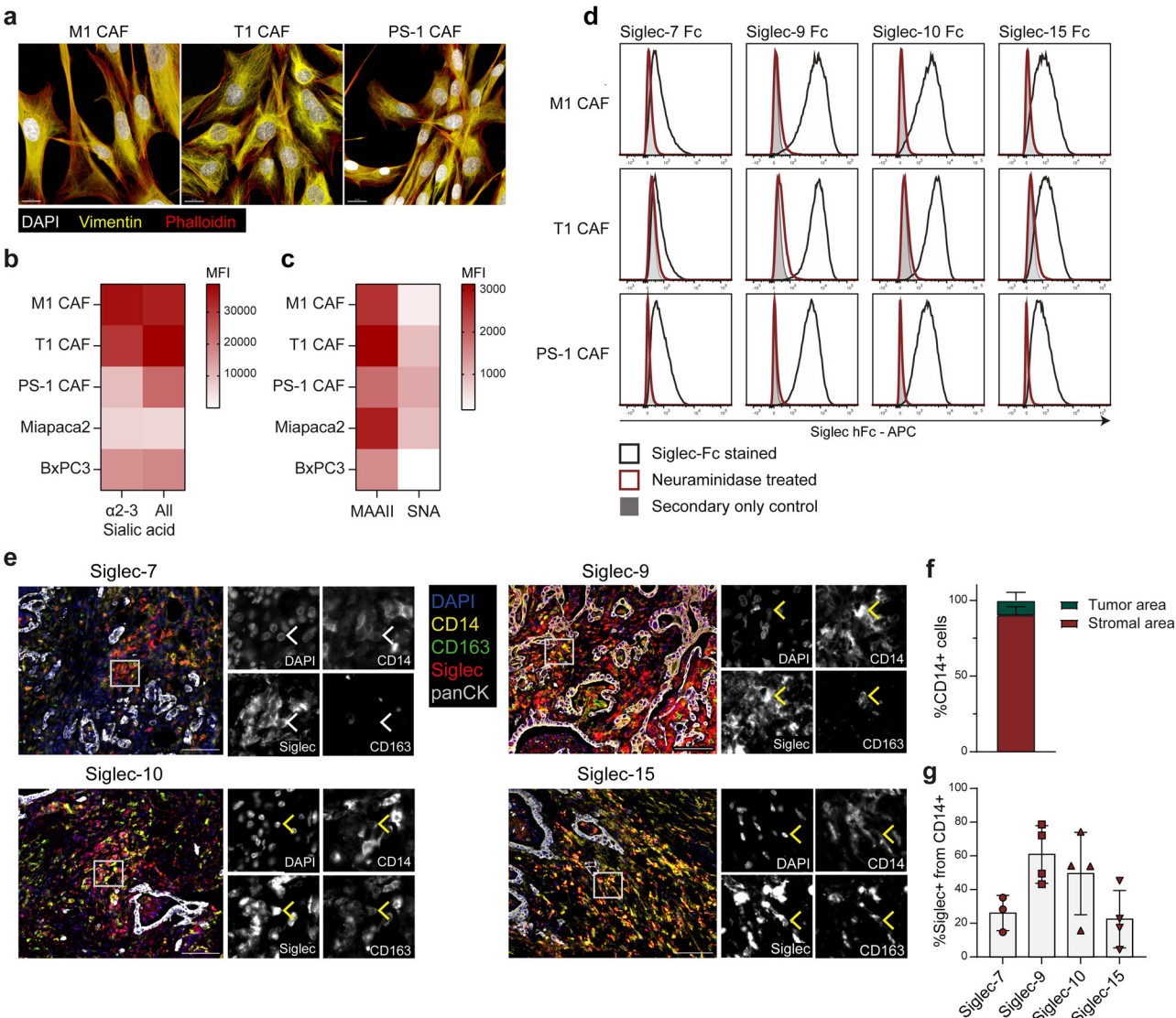

**Fig. 2 | Sialic acids on CAFs serve as ligands for Siglec-7, -9, -10 and -15 present on myeloid cells in PDAC. a** Morphology of CAF cell lines by immunofluorescence staining. Scale bar equals 20 μm. **b** Sialic acid expression on tumor and CAF cell lines assessed by flow cytometry using Lectenz probes. **c** Sialic acid expression on tumor and CAF cell lines assessed by flow cytometry using the plant lectins MAAII (binding α2-3 linked sialic acids) and SNA (binding α2-6 linked sialic acids). **d** Flow cytometric analysis of Siglec ligand expression in CAF cell lines treated with/without neuraminidase. **e** Multiplex immunohistochemistry on a PDAC biopsy for myeloid cells (CD14), TAMs (CD163), Siglec receptors and tumor cells (panCK). White boxes indicate zoomed in areas for which individual channels are shown in black/white. Image is representative for ≥3 patients. Scale bar equals 100 μm, white arrows indicate CD14+ Siglec+ cells, yellow arrows indicate CD14+ CD163+ Siglec+ cells. **f** Quantification of CD14+ cells in tumor region or stromal region of determined by panCK signal. **g** Percentage Siglec+ cells from total CD14+ cells in the stroma.

encompassing *O*-glycosylation, *N*-glycosylation, and glycosphingolipids glycosylation (Fig. 3a, Supplementary Data 1). *O*- and *N*-glycosylation is the process of covalently attaching glycans to proteins at serine/threonine or asparagine residues, respectively[59,60]. Glycosphingolipids on the other hand, are the major class of glycolipids[61]. This analysis revealed that the most abundantly expressed glycans were present in all CAF cell lines, and include gangliosides GM3 and GM2, several sialylated complex *N*-glycans, and the *O*-glycans disialyl-T and sialyl-T antigen (also called (di)sialyl core 1). However, notable differences were observed, particularly in the PS-1 CAFs that exhibited a different profile than the M1 and T1 CAFs. The PS-1 CAFs showed a restricted pattern in glycosphingolipids, including only GM3 and GM2 gangliosides. In contrast, the M1 and T1 CAFs displayed a more diverse ganglioside pattern, expressing additional gangliosides such as GM1 and GD1a, along with sialylated (neo-) lacto series glycosphingolipids (nsGSLs) (Fig. 3a, Supplementary Data 1). Similarly, the *N*-glycosylation and *O*-glycosylation patterns for PS-1 CAFs were more limited than those for M1 and T1 CAFs (Fig. 3a, Supplementary Data 1).

To determine whether the Siglec ligands on CAFs differ from those on tumor cells, the cell lines were treated with inhibitors for *O*-glycosylation, *N*-glycosylation, and glycosphingolipid synthesis. We previously reported that Siglec-7 ligands on tumor cells are mainly expressed on *O*-glycosylated proteins. Similarly, treating CAFs with an *O*-glycosylation inhibitor significantly reduced Siglec-7 ligands (Supplementary Fig. 3b). In contrast, while the *O*-glycosylation inhibitor significantly reduced Siglec-9 ligands on BxPC3 cells, PS-1 CAFs exhibited a significant reduction in response to the N-glycosylation inhibitor (Fig. 3b). The Siglec-9 ligands on M1 CAFs and T1 CAFs were reduced in response to both *O*-glycosylation and *N*-glycosylation inhibitors, although this reduction was modest and did not reach statistical significance (Fig. 3b). These results indicate that the Siglec-9 ligands on CAFs, and particularly those on PS-1 CAFs, differ from those on tumor cells. Furthermore, these results suggest variations in Siglec-9 ligands between PS-1 CAFs and M1 and T1 CAFs, with redundancy of Siglec-9 ligands in M1 and T1 CAFs.

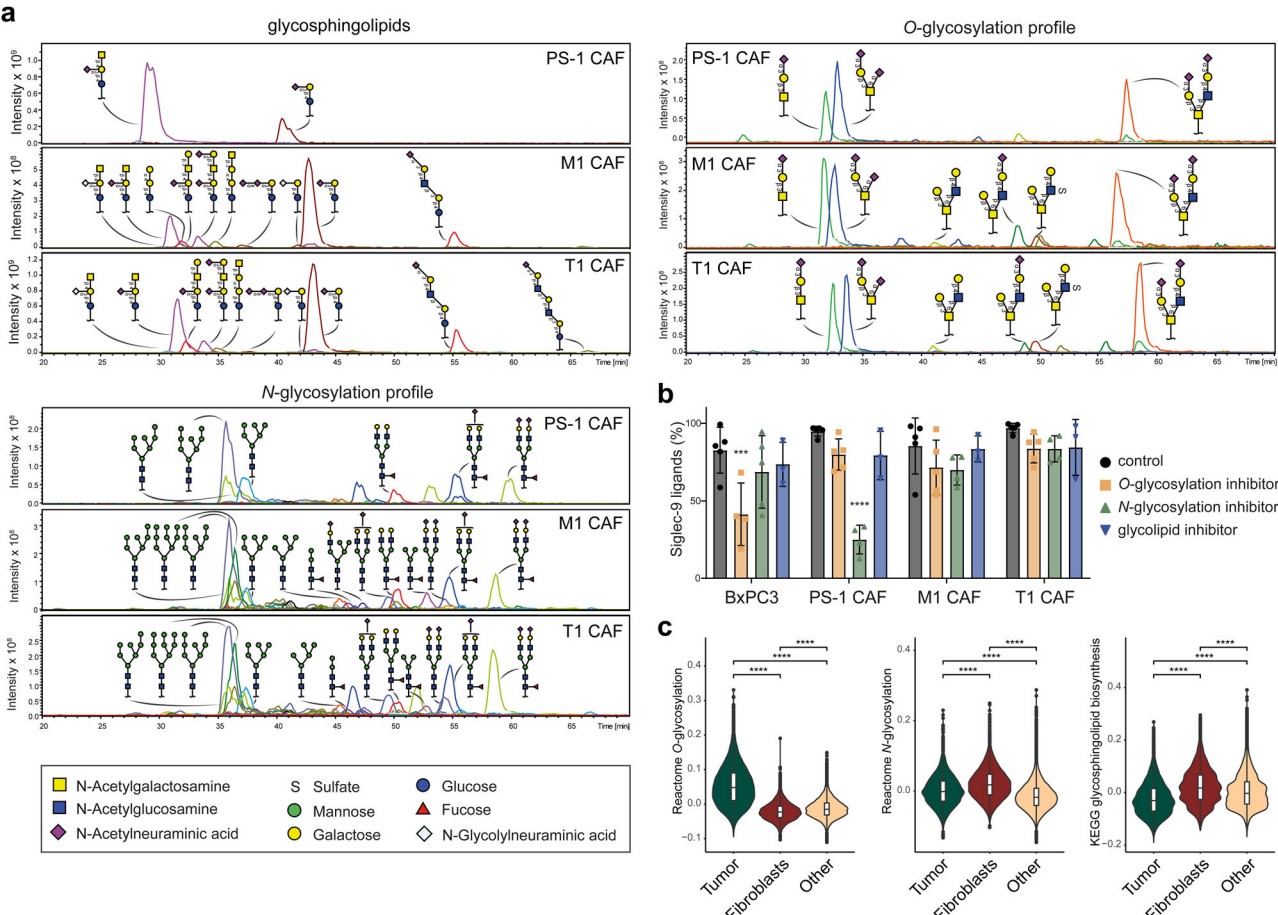

**Fig. 3 | Detailed glycan profiles of CAF cell lines. a** Glycosphingolipid, *N*-glycan and *O*-glycan profiles of CAF cell lines determined by mass spectrometry (*n* = 3). **b** Percentage of cells expressing Siglec-9 ligands after treatment with glycosylation inhibitors for glycosphingolipid synthesis (PPMP), *N*-glycosylation (Kifunensine) and *O*-glycosylation (Benzyl-GalNAc). Statistical analysis using two-way ANOVA and dunnett's multiple comparison test, comparing individual glycan inhibitor conditions to control within each cell line. **c** Gene module scores for *O*-glycosylation, *N*-glycosylation and glycosphingolipid biosynthesis in tumor cells, fibroblasts and other cells using scRNA-seq dataset from Peng et al.[50].

To further investigate the glycosylation pathways in tumor cells and CAFs, we analyzed the gene scores for *O*-glycosylation, *N*-glycosylation and glycosphingolipids glycosylation synthesis pathways within the scRNA-seq dataset from Peng et al.[50] (Fig. 1f). This analysis revealed that tumor cells exhibited a significantly enriched *O*-glycosylation score, while fibroblasts demonstrated a significantly higher *N*-glycosylation and glycosphingolipids glycosylation synthesis score (Fig. 3c). These results suggest that *N*-linked glycans and glycosphingolipid glycans are the most important sialic acid-containing glycoconjugates in CAFs.

### CAFs differentiate monocytes to immunosuppressive macrophages resembling TAMs

Infiltrating monocytes are a source of TAMs in PDAC[62]. We have previously shown that tumor cells can differentiate monocytes to TAMs[36]. To evaluate whether CAFs may also contribute to monocyte to TAM differentiation, we co-cultured a panel of fibroblasts and tumor cells with human monocytes isolated from fresh PBMCs and compared their phenotype using specific macrophage markers with flow cytometry (Fig. 4a). As controls, monocytes were differentiated into moMACs and moDCs as described above, or co-cultured with stellate cells isolated from pancreatitis (iHPSC). To analyse the phenotype of the differentiated monocytes we pre-gated on CD45$^+$ cells and performed dimensional reduction analysis (Fig. 4b). The tSNE presented distinct clusters of M1- and M2-moMACs and moDCs, while the macrophages from the co-culture with fibroblasts or PDAC tumor cell lines clustered together between monocytes and M-CSF-differentiated moMACs

(Fig. 4b). The co-clustering of monocytes co-cultured with PDAC tumor cells or fibroblasts, indicates that fibroblasts can also differentiate monocytes to macrophages with a TAM phenotype, characterized by expression of CD163 and CD206, markers known to be associated to pro-tumoral immunosuppressive TAMs[63,64] (Fig. 4b, c, Supplementary Fig. 4a). Surprisingly, CAFs induced significantly more differentiation to CD163$^+$CD206$^+$ TAMs than PDAC tumor cells (BxPC3) (Fig. 4d). Stellate cells isolated from pancreatitis (iHPSC) did not show this enhanced capacity to differentiate monocytes to CD163$^+$CD206$^+$ macrophages (Fig. 4d). The co-culture of CAFs and monocytes also contained increased IL-10 levels (Fig. 4e), and compared to the co-culture with PDAC tumor cells, the macrophages showed a trend toward enhanced expression of PD-L1 (Fig. 4f). In addition, CAF-induced TAMs express CD86 and HLA-DR (Supplementary Fig. 4b).

We also analyzed Siglec expression on TAMs after the co-culture with tumor cells or CAFs. All co-cultures resulted in TAMs expressing Siglec-7 and -9 receptors but showed minimal expression of Siglec-10 (Fig. 4g). Siglec-15 was expressed on approximately 50% of the CAF-differentiated TAMs (Fig. 4g).

To further investigate the functional characteristics of CAF-differentiated TAMs, we differentiated monocytes in the presence of a CAF-conditioned medium (CM) and co-cultured the macrophages with autologous CD8$^+$ T cells. Addition of CAF-CM during monocyte differentiation significantly increased the percentage of CD163$^{high}$CD206$^{high}$ macrophages, indicating that secreted product from CAF can drive TAM

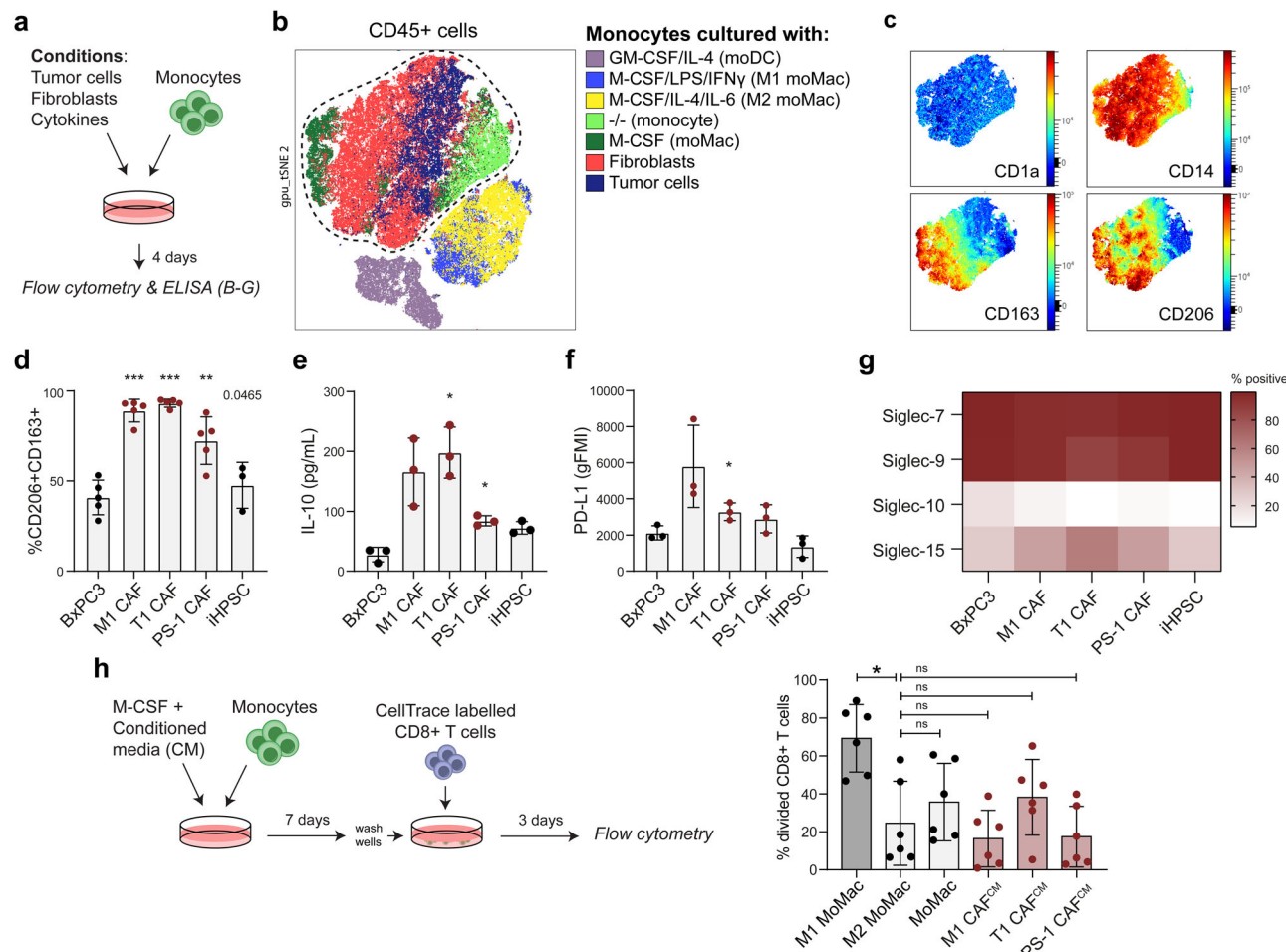

**Fig. 4 | Compared to tumor cells, PDAC CAFS have enhanced capacity to differentiate monocytes to macrophages with an immune suppressive phenotype.** **a** Schematic representation of the experimental setup related to Figure (**b–g**). Monocytes were co-cultured with either a fibroblast cell line (including M1 CAF, T1 CAF, PS-1 or iHPSC), or with a tumor cell line. **b** tSNE of monocyte-derived immune cells after co-culture with either fibroblasts or tumor cell lines. Additionally, tSNE shows monocytes differentiated in vitro using cytokines which serve as controls. Cells plotted in tSNE are pre-gated CD45+CD14+ cells, with the exception of the moDCs that were pre-gated only for CD45+ cells. **c** Expression of macrophage markers in the cluster containing TAMs and fibroblast-induced moMACs. **d** Percentages CD206+CD163+ cells gated from CD45+CD14+ cells. Statistical analysis with paired $t$ test compared to BxPC3. Statistical significance from $p \leq 0.0125$ (Bonferroni correction). **e** Cytokine levels of IL-10 in the co-cultures. Statistical analysis with one-way repeated measures ANOVA and dunnett's multiple comparison test, comparing fibroblast cell lines to BxPC3. **f** PD-L1 expression on differentiated monocytes after co-culture with BxPC3 or fibroblast cell lines. Statistical analysis with one-way repeated measures ANOVA and dunnett's multiple comparison test, comparing fibroblast cell lines to BxPC3. **g** Percentage of Siglec positive cells after co-culture with BxPC3 or fibroblast cell lines, gated from CD45+CD14+. Data represents the mean of 3 donors. **h** CD8+ T cell proliferation after 3 days of co-culture with different macrophage phenotypes. Macrophage phenotypes were generated by differentiating monocytes in the presence of conditioned media from M1 CAF, T1 CAF or PS-1 CAF, or alternatively by differentiating monocytes with M-CSF (moMac), or differentiating and polarizing them with IFN-γ (M1-moMac) or IL-4/IL-6 (M2-moMac). Statistical analysis with one-way repeated measures ANOVA and dunnett's multiple comparison test, comparing conditions to M2-moMAC control.

differentiation (Supplementary Fig. 4c). Furthermore, while the control M1-moMACs were potent stimulators of CD8+ T cell proliferation, CAF-conditioned macrophages induced T cell proliferation to a similar extent as M2-moMACs (Fig. 4h). These results demonstrate that CAFs are potent inducers of monocyte-to-macrophage differentiation, driving their differentiation toward an immunosuppressive phenotype.

## CAF-derived sialic acids instruct TAM differentiation via Siglec-9

Macrophage differentiation and polarization towards a tumor-promoting phenotype involve multifaceted processes influenced by various factors. Sialic acid-containing glycans, known regulators of macrophage behavior through interactions with Siglec receptors, contribute to the instruction of immunosuppressive macrophages[36,39]. Given the prevalence of sialylated glycans in CAFs, we hypothesized a role for the sialic acid-Siglec axis in CAF-mediated TAM differentiation. Underscoring the relevance of N-glycosylation in CAFs (Fig. 3c), our focus directed towards exploring the

impact of CAF sialylation on TAM differentiation in the context of PS-1 CAFs, where N-glycans play a significant role in Siglec-9 binding.

To understand how the sialic acid-Siglec axis is involved in PS-1 CAF-mediated TAM differentiation, we first interfered with the overall sialylation through treatment with a sialyltransferase inhibitor (SI), which led to a substantial reduction in sialic acid positive cells without affecting cell viability (i.e., from 80 to 100% to <10% positive cells, Supplementary Fig. 5a, b). To use the SI in a co-culture setting, the SI should be washed away before addition of monocytes to prevent a direct effect of SI on the monocytes. Removal of SI from the culture led to partial recovery of sialic acid expression after 3 days in PS-1 CAFs (Supplementary Fig. 5c). Next, SI-treated PS-1 CAFs were co-cultured with monocytes to investigate the role of CAF-derived sialic acids in TAM differentiation (Fig. 5a). Abrogation of sialic acid expression in the PS-1 CAFs significantly reduced the differentiation to CD163+CD206+ TAMs, and increased the inflammatory marker CD86 (Fig. 5a, b). As a second approach to study the role of CAF

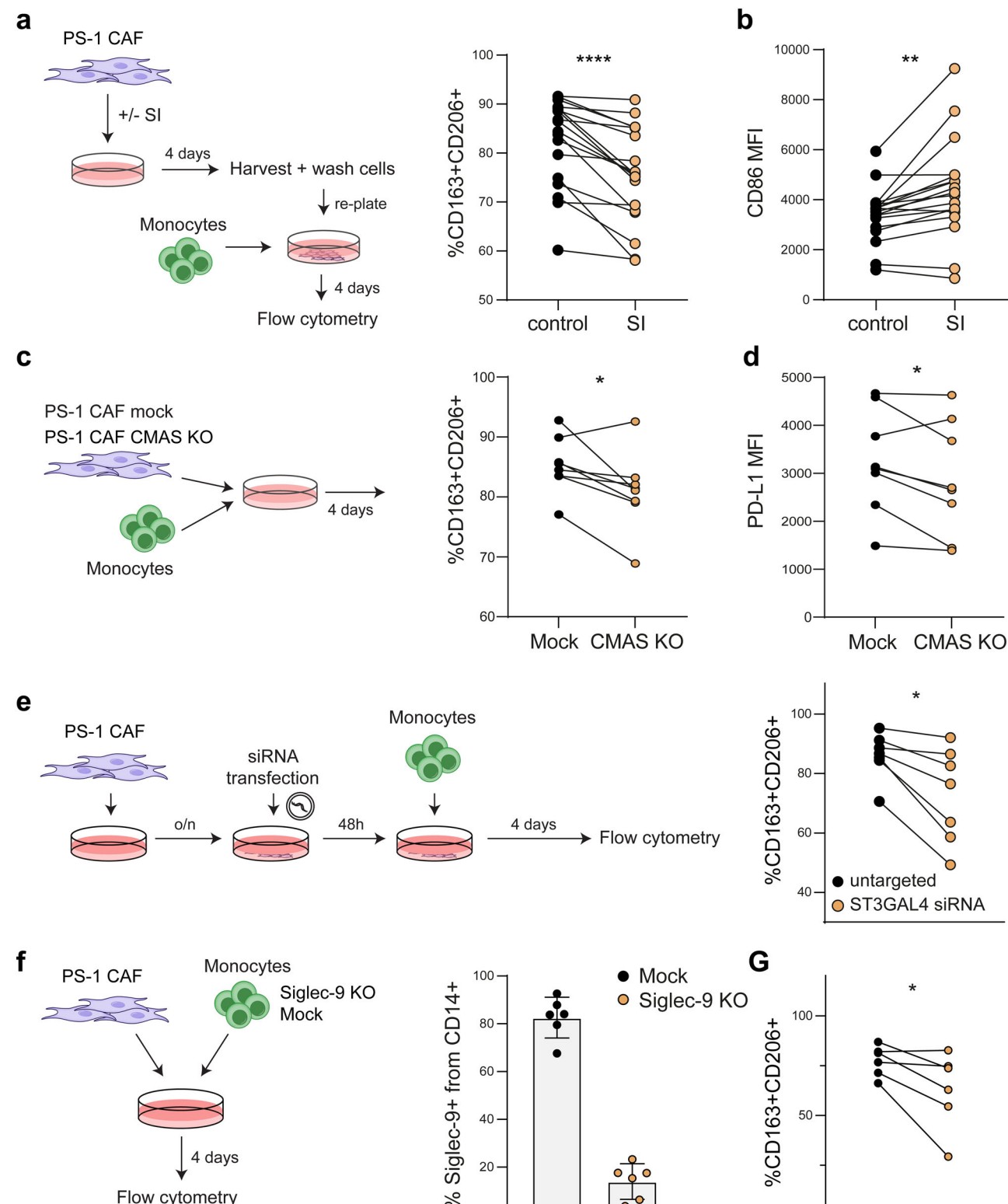

**Fig. 5 | Removal of sialic acids on PS-1 CAFs results in reduced TAM differentiation. a** Percentage of CD163+CD206+ cells within the CD45+CD14+ population after co-culture with PS-1 CAFs, treated prior to the co-culture with SI or DMSO as control. **b** Expression of CD86 in macrophages after co-culture with PS-1 CAFs, treated prior to the co-culture with SI or DMSO as control. **c** Differentiation of monocytes after co-culture with PS-1 CAF Mock or CMAS KO. **d** PD-L1 expression on macrophages after co-culture with PS-1 CAF Mock or CMAS KO. **e** Percentage of CD163+CD206+ cells within the CD45+CD14+ population after co-culture with PS-1 CAF, transfected with untargeted or ST3GAL4 siRNA. **f** Percentage of Siglec-9+ cells within CD14+ monocytes on day 4 of co-culture after transfection with Siglec-9 KO plasmid or Mock plasmid. **G** Differentiation of Mock and Siglec-9 KO monocytes after co-culture with PS-1 CAF. %CD163+CD206+ cells were gated within CD14+ cells for Mock transfected monocytes and gated on the CD14+ Siglec-9- population for the Siglec-9 KO transfected monocytes.

sialic acids on TAM differentiation, we generated a CMAS KO cell line to deplete sialic acids from the cell surface. The CMAS enzyme is responsible for generating the activated sialic acid sugar donor, which is subsequently transported into the Golgi for glycoprotein attachment (Fig. 1c). CMAS KO led to complete depletion of sialic acids from the surface of the PS-1 CAFs, including all the ligands for Siglec-7, -9, -10 and -15 (Supplementary Fig. 5f). In line with the effect of the SI, removal of sialic acids after knocking out the CMAS enzyme in the PS-1 CAFs decreased the differentiation to CD163+CD206+ TAMs (Fig. 5c). In addition, removal of CAF sialic acids reduced PD-L1 and HLA-DR expression on TAMs (Fig. 5d, Supplementary Fig. 5g). Together, these data show that sialic acids on CAFs contribute to TAM differentiation in the context of the PS-1 CAFs.

We identified ST3GAL4 as the sialyltransferase enzyme associated with CAFs (Fig. 1g, i). To study whether ST3GAL4 plays a role in CAF-mediated TAM differentiation, we knocked down ST3GAL4 using siRNA. In line with previous results, ST3GAL4 knockdown in PS-1 CAFs resulted in reduced TAM differentiation (Fig. 5e). Given that ST3GAL4 is involved in generating sialylated glycan ligands for Siglec-9[36,65], we knocked-out Siglec-9 in primary CD14+ monocytes to assess whether this receptor contributes to the CAF-mediated TAM differentiation. Siglec-9 was significantly reduced in CD14+ macrophages following transfection with the Siglec-9 KO plasmid, without affecting monocyte viability (Fig. 5f, Supplementary Fig. 5j). Interestingly, Siglec-9 KO in monocytes reduced CAF-mediated TAM differentiation in co-culture with PS-1 CAFs (Fig. 5G). The PS-1 CAFs mediated this effect, as Siglec-9 KO did not affect cytokine-induced differentiation of moMACs (Supplementary Fig. 5k). Thus, our results indicate that Siglec-9 is responsible for sensing the CAF sialic acids and mediates the CAF-driven TAM differentiation in the context of the PS-1 CAFs.

The effect of M1 and T1 CAF sialylation on TAM differentiation was also analyzed, using SI treatment, siRNA-mediated KD of ST3GAL4, and in Siglec-9 KO monocytes. Technical challenges prevented the generation of CMAS KO in M1 and T1 CAFs. Sialylinhibitor treatment ablated sialic acids from the M1 and T1 CAFs, without compromising cell viability, an effect retained in the T1 CAFs after SI removal (Supplementary Fig. 5a–c). SI treatment on M1 CAFs did not influence CAF-driven TAM differentiation (Supplementary Fig. 5d). However, sialic acid removal from M1 CAFs led to

reduced CD163 expression on TAMs, suggesting a nuanced effect of M1 CAF sialylation on TAM differentiation (Supplementary Fig. 5e). Conversely, in T1 CAFs, SI treatment increased differentiation towards CD163+CD206+ TAMs (Supplementary Fig. 5d). Neither ST3GAL4 knockdown in CAFs nor Siglec-9 KO in monocytes affected the differentiation to CD163+CD206+ TAMs in co-cultures with M1 or T1 CAFs (Supplementary Fig. 5i, m). These results contrast with the impact of PS-1 CAF sialylation on TAM differentiation, highlighting the context-dependent role of CAF sialylation in differentiating monocytes towards CD163+CD206+ TAMs. The diversity in sialylated glycan profiles between CAF cell lines (Fig. 3a, b) may underly the differential biological effects of CAF glycosylation on monocyte differentiation.

## Relative influence of tumor cells and CAFs in sialic acid-mediated TAM differentiation

Given that PS-1 CAF-derived sialic acids contributed to TAM differentiation similarly to tumor sialylation[36], we investigated the individual contributions of tumor- and CAF-derived sialylation in this process. In a co-culture setup of BxPC3 and PS-1 CAFs at a 1:4 ratio, BxPC3 cells proliferated faster, forming islands surrounded by PS-1 CAFs after 4 days (Fig. 6a). Before co-culturing BxPC3, PS-1 CAFs and monocytes, tumor cells and CAFs were treated separately with SI or vehicle (DMSO). Interestingly, when exclusively pre-treating PS-1 CAFs with SI before co-culture, a trend towards diminished differentiation into CD163+CD206+ TAMs emerged, accompanied by increased expression of CD86, and a trend towards elevated HLA-DR levels (Fig. 6b). Surprisingly, exclusive pre-treatment of BxPC3 with SI did not impact the differentiation into CD163+CD206+ TAMs (Fig. 6b), despite SI treatment of BxPC3 significantly reducing the expression of CD163, CD206, HLA-DR, and PD-L1 on monocytes when PS-1 CAFs were absent in the co-culture (Supplementary Fig. 6a). Only when both BxPC3 and PS-1 CAFs were treated with SI, a significant reduction in TAM differentiation was observed, and the TAMs expressed increased CD86 and HLA-DR levels (Fig. 6a). These results demonstrate the essential requirement for losing both tumor and CAF sialylation to reduce TAM differentiation. Importantly, these findings indicate a more prominent role for CAF sialylation in this process.

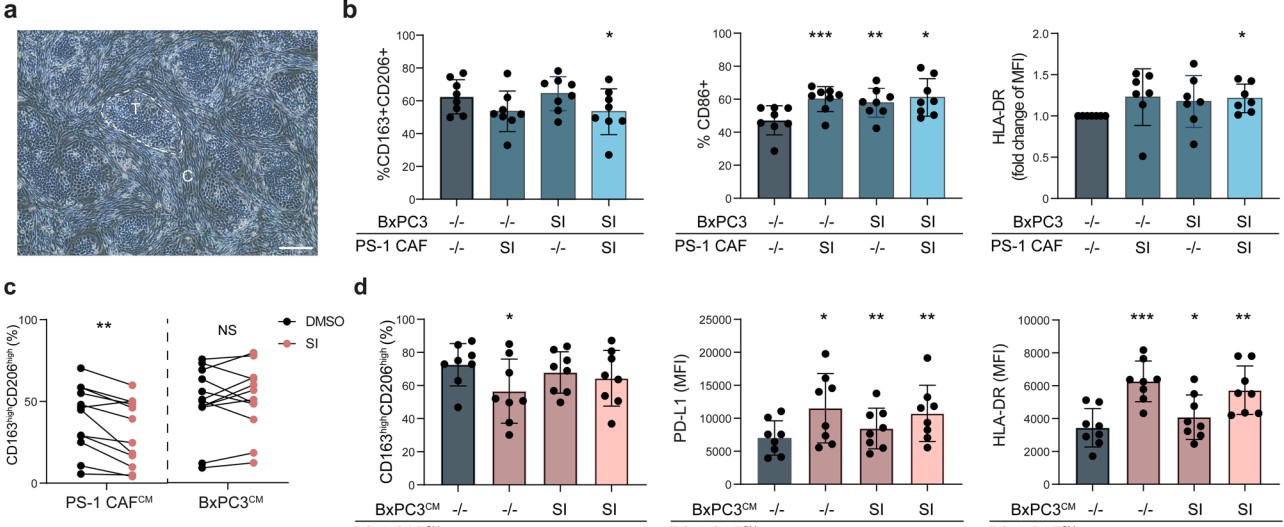

**Fig. 6 | Both tumor and stromal sialylation involved in TAM differentiation. a** Brightfield image of BxPC3 and PS-1 after 4 day co-culture. Dotted line illustrates an island of BxPC3 tumor cells "T", surrounded with PS-1 cells marked with "C". Scale bar equals 200 μm. **b** Percentage of CD163+CD206+ cells, CD86+ cells and MFI of HLA-DR within the CD45+CD14+ population after co-culture with BxPC3 and PS-1, that were treated prior to the co-culture with SI or DMSO as control. Statistical analysis with one-way repeated measures ANOVA and dunnett's multiple comparison test. **c** Percentage of CD163+CD206+ cells within the CD45+CD14+

population after differentiation in the presence of conditioned media from either BxPC3 or PS-1. Conditioned media was generated over 24 h, after 3-day treatment of cells with SI or DMSO as control. **d** Percentage of CD163+CD206+ cells and MFI of PD-L1 and HLA-DR, within the CD45+CD14+ population after differentiation in the presence of conditioned media from BxPC3 and PS-1. Conditioned media was generated over 24 h, after 3-day treatment of cells with SI or DMSO as control. Statistical analysis with one-way repeated measures ANOVA and dunnett's multiple comparison test.

In an alternative approach to discern the relative contributions of tumor and CAF sialylation, monocyte differentiation was induced in the presence of CM obtained from BxPC3 tumor cells or PS-1 CAFs treated with vehicle control (PS-1$^{CM-DMSO}$ and BxPC3$^{CM-DMSO}$) or treated with SI (PS-1$^{CM-SI}$ and BxPC3$^{CM-SI}$). This approach prevented potential confounding effects of proliferation differences between BxPC3 and PS-1 CAFs. TAM differentiation was reduced in the presence of PS-1$^{CM-SI}$, but not with BxPC3$^{CM-SI}$, compared to control (Fig. 6c). The expression of HLA-DR and PD-L1 on TAMs increased when differentiated in the presence of PS-1$^{CM-SI}$ or BxPC3$^{CM-SI}$ (Supplementary Fig. 6b, c). These findings implicate that secreted sialylated products from CAFs, more so than from tumor cells, are involved in TAM differentiation. Subsequently, equal amounts of PS-1$^{CM}$ and BxPC3$^{CM}$ were added simultaneously. When monocytes were differentiated in the presence of both PS-1$^{CM}$ and BxPC3$^{CM}$ in equal ratio, derived from SI or vehicle-treated cells, a reduction in differentiation towards CD163$^{high}$CD206$^{high}$ TAMs was observed only with PS-1$^{CM-SI}$ (Fig. 6d). Additionally, the largest increase in PD-L1 and HLA-DR expression was induced by PS-1$^{CM-SI}$ (Fig. 6d). These results suggest that while tumor and CAF sialylation are essential for contact-dependent TAM differentiation, CAF-derived sialylation plays a more dominant role in TAM differentiation through secreted products.

## Discussion

This study aimed to evaluate the sialic acid expression in CAFs and its role in immune modulation. Our results demonstrate elevated sialic acid levels in CAFs compared to tumor cells, with ST3GAL4 identified as the key regulatory enzyme in CAF sialylation. Unlike PDAC tumor cell sialic acids binding Siglec-7 and -9[36], sialic acids on CAFs serve as ligands for Siglec-7, -9, -10 and -15. Importantly, we show that CAF-derived sialic acids influence the differentiation of monocytes to CD163$^+$CD206$^+$ macrophages, a phenotype associated with tumor-promoting immunosuppressive TAMs. This work uncovers CAF-immune crosstalk dependent on CAF glycosylation and its interaction with suppressive receptors on immune cells.

We show that CAF-derived sialic acids interacted with the immune suppressive receptors Siglec-7, -9, -10 and -15. Of these four Siglecs, Siglec-9 was most abundant within the PDAC TME and played a significant role in monocyte-to-TAM differentiation by CAFs. Yet, we cannot exclude the involvement of other Siglec receptors in this process. Siglec-7, -10 and -15 can also modulate myeloid cells in cancer. Siglec-10 binds CD24 on breast cancer cells and prevents macrophage-mediated phagocytosis, indicating that Siglec-10 may also suppress macrophage functioning[37]. Siglec-7 is involved in TAM differentiation by PDAC tumor cells and Siglec-15 has been described as target for normalizing immunotherapy, with its expression on macrophages and its link with suppressing T cell proliferation[36,38]. We found that blood-isolated monocytes did not express Siglec-15, but gained Siglec-15 expression upon differentiation to M0- or M2-moMACs in vitro. We therefore speculate that Siglec-15 could play a role in macrophage polarization at later stages rather than in the early steps of macrophage differentiation.

Our data shows that PS-1 CAF-derived sialic acids contribute to TAM differentiation and polarization towards an immunosuppressive phenotype, at least in part via Siglec-9. These results align with previous reports showing that engagement of Siglec-9 by sialylated dendrimers, or by sialylated MUC1 induces immunosuppressive macrophages with a TAM-like phenotype[36,39]. In addition, PDAC tumor sialylation stimulates TAM differentiation via Siglec-7 and Siglec-9[36]. Even though similar effects on TAM differentiation can be observed between these studies, there is a key difference in the Siglec ligands between tumor cells and CAFs (Fig. 3b). Importantly, loss of both tumor and CAF sialylation was required to reduce TAM differentiation. We recently showed that ablation of sialic acid on PDAC tumor cells in vivo reversed the T cell excluded phenotype and synergized with immunotherapy[66]. Loss of sialic acid on tumor cells alone did not impact tumor growth in this mouse model, and mice were not cured with immunotherapy[66]. Therefore, removing both tumor and CAF sialylation

may be necessary to overcome the barriers for immunotherapy efficacy in PDAC.

We observed contrasting effects of sialylation on TAM differentiation among the three CAF cell lines. While SI treatment of PS-1 CAFs reduced TAM differentiation, it enhanced TAM differentiation in context of T1 CAFs. Their glycosylation profiles exhibited important differences, particularly in PS-1 cells compared to M1 and T1 CAFs. Siglec-9 binding to glycans also differed, with N-glycosylated proteins being the primary ligands in PS-1 cells, and ligands in M1 and T1 CAF cells being modestly reduced by both O- and N-glycan inhibitors (Fig. 3b). These findings suggest that the specific context of glycoproteins may influence the biological outcome of Siglec interaction. Supporting this notion, we observed differential expression of several highly glycosylated proteins in the cell lines (PDPN, CD106, CD146, Supplementary Fig. 2f). Further research is needed to investigate specific Siglec glycoprotein ligands on CAFs and to understand the contexts in which CAF sialylation promotes or inhibits immunosuppression.

A limitation of the current study is the use of immortalized CAF cell lines in vitro, to study the biological implications of CAF sialylation on myeloid cells. While some CAF characteristics are maintained in culture, some traits are likely not. Although the PS-1 are activated, lost their stellate cell quiescent phenotype and express several CAF markers and characteristics, they are originally isolated from heatlhy pancreas and may not fully recapitulate CAFs in vivo. Furthermore, cell lines will not represent the full heterogeneity and plasticity observed in vivo. All three CAF cell lines displayed markers of myCAFs, including α-SMA, but also secreted IL-6, associated with an iCAF phenotype[53]. These data underscore the challenge of distinguishing between iCAF and myCAF phenotypes in human fibroblasts in vitro, implicating either the absence of definitive markers or the inability of in vitro fibroblasts to recapitulate the iCAF/myCAF distinction observed in transcriptomic data. To capture the full complexity and heterogeneity of CAF subsets and their interaction with immune cells, future studies should evaluate the role of CAF-derived sialic acids on tumor progression and immune modulation in vivo.

Although our study focused on the effect of CAF sialylation on macrophage during their differentiation and polarization, sialic acids can also modify other immune cells through Siglec engagement. A recent study, in particular, demonstrated that stromal sialylation can impede T cell proliferation[67]. Furthermore, the engagement of Siglec-9 on CD8$^+$ T cells has been shown to diminish TCR signaling and effector function[56,57]. Sialic acids can also affect effector functions of DCs by impacting their maturation, cross-presentation and T-cell priming ability[68–71].

Compared to other cell types in the PDAC TME, the sialyltransferase enzyme ST3GAL4 was highly expressed in CAFs. ST3GAL4 is involved in the synthesis of Siglec-9 ligands on PDAC tumor cells and is associated with decreased survival[36,65]. In addition, ST3GAL4 is related to an increased invasive phenotype in tumor cells as it generates the glycan sialyl-Lewis X, which facilitates cell adhesion[72,73]. Interestingly, recent studies also observed increased ST3GAL4 expression in the stroma in other cancer types, including colorectal, lung, cervical and esophageal cancer[67,74]. Stromal sialylation therefore likely plays a role in other cancer types as well.

Over the past decade, multiple strategies have been developed to target the sialic acid-Siglec axis, including tumor-targeted degradation of sialic acids, sialyltransferase inhibitors, and Siglec blocking antibodies[38,45,46,75]. Blocking Siglec-7, -9, -10, -15 simultaneously on monocytes is challenging, and commercially available blocking antibodies do not sufficiently block these receptors. Given the potential involvement of multiple Siglec receptors in both tumor and CAF immune crosstalk, interference with the ligand, sialic acid, would be the preferred approach, which we accomplished in CAFs by treatment with a sialyltransferase inhibitor or genetic KO of sialylation enzymes. Interestingly, a phase 1/2 clinical trial, which includes PDAC patients, is currently investigating the safety and potential of sialidase treatment (NCT05259696)[76]. An intriguing avenue for further investigation is the potential impact of this treatment on CAF sialylation.

In conclusion, CAF sialic acids form ligands for multiple Siglec receptors, such as Siglec-7, -9, -10 and -15 on immune cells and can modulate TAM differentiation. Therapeutic interventions targeting the sialic acid-Siglec axis are currently focussed on tumor sialylation. We propose that CAF sialylation should also be considered in the development of sialic acid-based therapies. Future research is necessary to reveal the role of CAF sialylation in modulation of other immune cells, such as NK cells, T cells and dendritic cells, and how CAF sialylation relates to tumor progression and immune evasion in vivo.

## Materials & methods

### Patient material
Formalin-fixed paraffin embedded (FFPE) tissue from PDAC patients was obtained from the Pathology Department of the Amsterdam UMC, location VUMC, with the approval from the Medical Ethical committee from the Amsterdam UMC, location VUMC. Written consent was obtained from all the patients. All ethical regulations relevant to human research participants were followed. Stage of lesions can be found in supplementary Table 1.

### Cell lines
For this project, several human fibroblast cell lines and PDAC tumor cell lines were used (Supplementary Fig. 2a). The fibroblasts M1 CAFs and T1 CAFs were generated in the laboratory of Dr. Prof. Tuveson and are derived from metastatic lung and primary human pancreatic ductal adenocarcinoma[14]. The human pancreatic stellate cell line PS-1 was a kind gift from Dr. Prof. H. Kocher[52]. PDAC tumor cell line MIA PaCa-2 was acquired via ATCC, BxPC3 was a kind gift from Dr. A. Frampton (Imperial College, London, UK). All cell lines were cultured in RPMI 1640 (Gibco) containing 10% Fetal Calf Serum (Biowest), 2 mM L-Glutamine (Gibco) and 1000 U per mL Penicillin-Streptomycin (Gibco), referred to as complete RPMI. All cell lines were routinely tested for mycoplasma using PCR.

### Collagen Gel contraction assay
Rat-tail collagen type I was reconstituted in 0.1% acetic acid (4 mg per ml). Cells were diluted in complete RPMI and seeded in the collagen solution at $2 \times 10^5$ cells per ml and 1 ml hydrogel was poured per well of 12-well plates. Hydrogels were polymerized for 2 h at 37 °C. Hydrogels were detached from the well surface to allow contraction and culture medium was added. Three times per week the culture medium was refreshed and pictures of gels were taken using a Sony WX500 camera during a total culture period of two weeks. Hydrogel surface area was measured using ImageJ software.

### Monocyte isolation and stimulation
Healthy donor buffy coats were collected by Sanquin, the Netherlands, from which peripheral blood mononuclear cells (PBMCs) were isolated by density gradient centrifugation with Ficoll-Paque PLUS (GE Healthcare). Using CD14 Microbeads (Miltenyi) human CD14$^+$ monocytes were isolated. Monocytes were cultured in RPMI 1640 (Gibco) containing 10% Fetal Calf Serum (Biowest), 2 mM L-Glutamine (Gibco) and 1000 U per mL Penicillin-Streptomycin (Gibco).

Monocytes were differentiated to monocyte-derived macrophages (moMACs) by the addition of 50 ng/mL M-CSF. To polarize moMACs, 10 ng/mL LPS, 20 ng/mL IFNγ, 20 ng/mL IL-4 and/or 20 ng/mL IL-6 was added on day 3. Monocyte-derived dendritic cells (moDCs) were generated with 20 ng/mL GM-CSF and 20 ng/mL IL-4.

Differentiation of monocytes in the presence of fibroblast-conditioned medium (CM) was done with 25 ng/mL M-CSF. Conditioned media was generated over a 24-hour period from cells treated with the Sialyltransferase Inhibitor 3Fax-Peracetyl Neu5Ac (SI) (Calbiochem, Sigma Aldrich). The cells underwent a 3-day incubation with 200 µM SI or vehicle (DMSO), followed by three washes with PBS before CM collection.

### Co-cultures
To study the effect of pancreatic fibroblast and tumor cell lines on monocyte differentiation, cell lines were plated in 24-well plates (50.000 cells per well) and co-cultured with monocytes (200.000 cells per well) for 4 days. After 4 days, supernatant was collected to analyze the cytokines with ELISA. Cells were harvested and analyzed by flow cytometry. Dimensional reduction analysis and visualization of co-cultures was done using OMIQ software from Dotmatics.

To evaluate the effect of macrophages on CD8$^+$ T cell proliferation, CD8$^+$ T cells were isolated from PBL's using negative isolation kit (Miltenyi, Cat#130-096-495), and frozen until use. After thawing, CD8$^+$ T cells were labeled with CellTrace™ Violet (Invitrogen, Cat#C34557), for 7 min at 37 °C while shaking. Autologous CD8$^+$ T cells were added to the macrophages in 4:1 ratio and cultured for 3 days in the presence of Ultra-LEAF™ Purified anti-human CD3 (clone OKT3, Biolegend).

In experiments using SI treatment, fibroblast cell lines were incubated with 200 µM SI or vehicle (DMSO) for 4 days, after which the cells were washed, re-plated and co-cultured with monocytes.

### CRISPR-cas9 gene knockout
Generation of a Siglec-9 knockout (KO) in monocytes was done using nucleofection of freshly isolated CD14$^+$ monocytes, as reported previously[77]. First, 6 µL of 160 µM crRNA (Dharmacon, CM-012842-01-0010) and 6 µL of 160 µM tracrRNA (Dharmacon, U-002005-50) were mixed and incubated at 37 °C to form 12 µL of gRNA:tracrRNA duplex specific for Siglec-9. Subsequently, 12 µL of 40 µM Cas9-NLS protein (Horizon Discoveries, CAS12206) was added, after which the sample was incubated for 15 minutes at 37 °C to form CRISPR-Cas9 ribonucleoproteins (crRNP). The crRNPs were stored at –70 °C until use. Freshly isolated CD14$^+$ monocytes were nucleofected with crRNP using the P3 Primary Cell 4D-Nucleofector™ X Kit L (Lonza, V4XP-3034) according to the manufacturer's protocol. The mock transfection was performed by exposure to the nucleofection process without crRNPs present. Per condition, 12.5 µL crRNP and $5 \times 10^6$ CD14$^+$ monocytes in 100 µL P3 Primary Cell Nucleofector™ solution were added to the Nucleovette™ vessel. After nucleofection (pulse code DK-100), cells were resuspended in pre-warmed complete RPMI and incubated for 30 minutes at 37 °C. After incubation, cells were harvested from the Nucleovette™ vessel, counted, and used for co-culture experiments.

The generation of CMAS KO in the PS-1 cell line was performed as reported previously[36]. Briefly, sgRNA strands for human *CMAS* gene (top strand CACCGATATCTGAACAGTGTAT; bottom strand AAACATAC ACTGTTCAGATATC.) were phosphorylated and annealed to clone it in the pSpCas9(BB)-2A-Puro plasmid, a gift from Feng Zhang (Addgene#62988). PS-1 cells were transfected using Lipofectamine™ LTX with PLUS™ Reagent (Invitrogen), selected with puromycin and sorted based on negative staining of α2-3-Lectenz (Lectenz Bio) using BD FAC-SAria™ Fusion FACS sorter. The control PS-1 mock was transfected with pSpCas9(BB)-2A-Puro plasmid without guide RNA.

### Flow cytometry
All stainings with plant lectins (Vector Laboratories), Lectenz (Lectenz Bio) or Siglec-Fc chimeras (R&D systems) were performed in HBSS containing magnesium and calcium (Gibco) supplemented with 0.5% fatty-acid free BSA (Sigma). Siglec-Fc chimeras were pre-incubated at 1 µg/mL with anti-human IgG Fc (Biolegend, clone: HP6017) for 15 minutes at room temperature, after which they were added to the cells. Similarly, 1 µg/mL lectins and Lectenz reagents were pre-incubated with streptavidin-APC before addition to the cells. Macrophages in Fig. 4 were analyzed with a 14-color antibody panel (supplementary Table 2) on the Cytek Aurora and analyzed with FlowJo v10 and OMIQ. Fibroblast markers PD-L1, HLA-DR and α-SMA were measured on Cytek Aurora, other experiments were analyzed using the Fortessa™ X-20 and analyzed with FlowJo v10 (list of antibodies in supplementary Table 2).

In indicated experiments, cells were treated at 37 °C for 30 minutes with neuraminidase from *Arthrobacter ureafaciens* (Roche Diagnostics, diluted 1:100). To assess the presence of sialylated structures in *N*-glycans, *O*-glycans, or glycolipids, cells were treated for 3 days with 10 µg/mL Kifunensine, 0.8 mM Benzyl-GalNAc, or 5 µM PPMP, respectively.

## Tissue stainings

Immunohistochemical (IHC) staining of tissues was performed on FFPE sections (5 μm). Tissue slides were stained with Hematoxylin & Eosin (HE) to verify tumor histology by a pathologist. After deparaffinization and antigen retrieval (DAKO, Tris-EDTA pH9 buffer), endogenous peroxidase activity was blocked by peroxidase-blocking solution (DAKO) and tissues were blocked with Carbo-Free Blocking Buffer (CFBB) (Vector Labs). Slides were stained with 2 μg/mL pan-Lectenz (Lectenz Bio), which was pre-complexed with 1 μg/mL streptavidin-HRP for 30 min before adding to the slides. For ST6GALNAC6 stainings, slides were incubated with ST6GAL-NAC6 antibody (Sigma Aldrich, HPA018890, 1:10) for two hours at 37 °C. After incubation with the primary antibody (Supplementary Table 2), slides were incubated with BrightVision Poly-HRP-Anti Mouse/Rabbit IgG Biotin-free (Immunologic, VWRKDPVO55HRP) for 30 minutes. IHC reactions were detected using DAB (3,3'-diaminobenzidine) and slides were counterstained with hematoxylin. After rehydration, slides were mounted with Entallan and scanned using Vectra Polaris (Akoya Biosciences). If applicable, neuraminidase from *Arthrobacter ureafaciens* (1:10 Roche Diagnostics) was applied after endogenous peroxidase blockade for one hour at 37 °C after which the protocol was continued.

Multiplex IHC stainings were performed using an Opal multiplex IHC kit (Akoya Biosciences, NEL821001KT), with detailed information about antibodies listed in supplementary Table 2. Staining was performed according to manufacturer's procedures. After staining, slides were counterstained with DAPI and mounted with Fluormount-G (ITK, 0100-01). Slides were scanned using the Vectra Polaris Automated Quantitative Pathology Imaging System (Akoya Biosciences), software version 1.0.13.

First, multiplex-stained slides were scanned using a ×20 magnification with multispectral slide scan bands in order to annotate regions for tumor content. These regions were imaged in a second round using a 40x magnification with multispectral field bands. Obtained multispectral images were acquired and unmixed using inForm® Tissue Analysis Software version 2.6.0 with a spectral library build using single stained samples.

Quantification of pan-Lectenz intensity was done in QuPath version 0.2.2[78]. After tissue segmentation between tumor and stroma was applied, annotations were exported to ImageJ version 1.53a and quantified for median intensity of pan-Lectenz staining. Image quantification of ST3GAL4 and Siglec expression was done using NIS-Elements (version 5.42.04). ST3GAL4 expression on stromal cells was quantified as ST3GAL4$^+$ cells in panCK- areas, lacking CD45.

## Microscopy of cell lines

For the immunofluorescent staining of fibroblasts, cell lines were grown on 8 well ibidi μ-slide (ibidi) for two days at 37 °C and 5% CO$_2$. After that time, medium was removed and cells were fixed with 4% paraformaldehyde (PFA) for 15 minutes at room temperature (RT), permeabilized with 0.1% Triton X-100 in PBS for 10 min at RT and blocked using 10% Normal mouse serum (NMS) in PBS containing 0.1% Tween 20. Cells were incubated overnight at 4 °C with anti-Vimentin Alexa Fluor 594 (Biolegend, dilution 1:200) in PBS containing 2% NMS and 0.1% Tween 20. Next, cells were washed four times with 0.1% Tween 20 in PBS and counterstained with Alexa Fluor 647 Phalloidin (Invitrogen, dilution 1:400) and DAPI (Invitrogen). Images were acquired using SP8 confocal microscope (Leica).

## Glycan profiling

Analysis of glycosphingolipid glycans, *N*-glycans and *O*-glycans of fibroblast cell lines was performed by PGC nano-LC-ESI-MS/MS in negative mode, as described previously[66].

## Cytokine analysis

Cytokines in the supernatant of cell lines were measured using LEGEN-Dplex Human Essential Immune Response Panel kit (Biolegend) and LEGENDplex Human HSC Myeloid Panel kit (Biolegend) according to manufacturer's instructions. Briefly, cell lines were plated and grown till 70% confluence, after which the culture media was refreshed. After 24 hours, cytokines were measured with the LEGENDPlex kits. IL-10 levels from co-cultures were measured using ELISA (supplementary Table 2).

## siRNA knockdown

The fibroblast cell lines were plated in a 6-wells plate and grown over night. Next, siRNA mediated knockdown of ST3GAL4 was achieved with DharmaFECT2 Transfection Reagent (Dharmacon) according to the manufacturer's instructions. A non-targeting siRNA (Dharmacon) was taken along as a control.

## Transcriptomic analysis

The single-cell RNA sequencing (scRNA-Seq) data previously published by Peng et al. was downloaded from the Genome Sequence Archive project PRJCA001063 as pre-processed row data and imported into the package *Seurat* (v4) for downstream analysis as described previously[36,50]. The dataset included tissues from 24 PDAC patients and 11 normal pancreas, for a total of 56601 cells. The function *FindMarkers* was used for the generation of gene sets corresponding to cancer cells and fibroblasts, selecting the significant genes that present a fold change equal or higher than 2 (Supplementary Table 3). For analysis of fibroblast subsets, clusters containing cells expressing the gene LUM were selected, renormalized using the function SCTransform (regressing out the percentage of mitochondrial genes) and batch effect was corrected using the package Harmony with default settings. Only samples containing 100 or more cells were used in this analysis, resulting in 6887 cells. We proceed to cluster cells using the functions FindNeighbors and FindClusters with the top 20 hermony dimensions and a resolution of 0.5. To clean the data, we removed cluster of cells based on the expression of markers specific for other cell types (PTPRC for immune cells, KRT19 for epithelial cells, or PRSS1 for acinar cells) or low-quality cells (defined by the high content of ribosomal and/or mitochondrial genes). This resulted in a total of 5344 fibroblasts, derived from 11 PDAC patients and 4 normal samples. For identification of CAF subtypes, cells were re-clustered as described before but using a resolution of 0.25. The function AddModuleScore was employed to generate gene scores for different glycosylation pathways using genesets previously described[74] (supplementary Table 3). The scripts used in this manuscript for the analysis of scRNA-Seq can be found in https://github.com/MolecularCellBiologyImmunology/Sialylation_CAF. Any additional information is available from the authors upon request.

The data from the PAAD project of the TCGA was obtained from the Broad Institute GDAC Firehose (https://gdac.broadinstitut.org)[79]. The package *GSVA* was used for the generation scores evaluating the expression of the different gene sets (corresponding sialylation pathways, fibroblasts and cancer cells, supplementary Table 3) and their association was evaluated using Spearman correlation.

Transcriptomic data from microdissected samples was downloaded from the NCBI Gene expression Omnibus (GEO) using the accession number GSE93326[49]. The package *limma* was used for the analysis of the differential gene expression, with FDR correction for multiple comparisons.

## Statistics and reproducibility

Statistical analysis was performed in Graphpad Prism 9.3.1. Comparison between two groups was done using the two-tailed paired Student $t$ test, unless stated otherwise in the Figure legend. A p-value of <0.05 was considered statistically significant (*$P \leq 0.05$, **$P \leq 0.01$, ***$P \leq 0.001$). All bars in graphs represent the mean and error bars represent the standard deviation (SD). All data points displayed in graphs in this paper represent biological replicates, indicated in Figure legends by $n$.

## Data availability

The single-cell RNA sequencing (scRNA-Seq) data from Peng et al. was downloaded from the Genome Sequence Archive project PRJCA001063[50]. The scripts used in this manuscript for the analysis of scRNA-Seq can be found in https://github.com/MolecularCellBiologyImmunology/Sialylation_CAF. The data from the PAAD project of the TCGA was obtained from the

Broad Institute GDAC Firehose (https://gdac.broadinstitut.org)[79]. Transcriptomic data from microdissected samples was downloaded from the NCBI Gene expression Omnibus (GEO) using the accession number GSE93326[49]. The remaining data are available within the Article, Supplementary files or available from the authors upon request. The source data are available in Supplementary Data 1.

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

## Acknowledgements

We would like to acknowledge the Microscopy and Cytometry Core Facility at the Amsterdam UMC-Location VUmc for providing assistance in cytometry experiments. This work is financially supported by KWF VU2014-7200 to K.B.; SPINOZANWO SPI-93-538 to K.B., E.R., T.E. and Y.K.; KWF 12789 to K.B., B.O.S., D.L.; LSH-TKI project DC4Balance LSM1806-SGF to AH and China Scholarship Council no. 202006940010 to C.L.

## Author contributions

K.B. and Y.v.K conceived the study. K.B., E.R., Z.H., C.L., D.W., B.O.S., K.O., T.v.E., A.d.H., J.P.N., D.A.T., E.G., M.F.B., M.W., S.J.v.V, Y.v.k designed experiments or provided essential technical support. K.B., E.R., Z.H., C.L., D.W., B.O.S., K.O., L.G., T.v.E., D.L., W.T., A.d.H., L.W., J.P.N.,A.M. and C.M.d.W. Carried out experiments and acquired experimental data. D.W. and M.W. performed glycan profiling. K.B., E.R., Z.H., C.L., D.W., B.O.S., K.O., T.v.E. and A.M. analyzed data. E.R. Performed transcriptomic analysis.

K.B. and Y.v.K drafted the manuscript. K.B., E.R., D.W., B.O.S., K.O., L.G., T.v.E., D.L., W.T., A.d.H., L.W., J.P.N., A.M., C.M.d.W., R.E.M., D.A.T., E.G., M.F.B., M.W., S.J.v.V and Y.v.K provided critical intellectual content. Y.v.K Supervised the study.

## Competing interests

The authors declare no competing interests

## Additional information

**Peer review information** : *Communications Biology* thanks Heinz Läubli, Julia M Houthuijzen and the other, anonymous, reviewer(s) for their contribution to the peer review of this work. Primary Handling Editor: Christina Karlsson Rosenthal. A peer review file is available.

