## [Peer review file · Communications Biology]

Reviewers' comments:

Reviewer #1 (Remarks to the Author):

Review report

Manuscript title: pancreatic cancer-associated fibroblasts modulate macrophage differentiation via sialic acid-Siglec interactions

Manuscript ID: COMMSBIO-23-0459-T

Reviewer: Dr. Julia Houthuijzen, Molecular Pathology, The Netherlands Cancer Institute.

In this manuscript the authors identified immune-modulatory effects of cancer-associated fibroblasts (CAFs) on macrophages via sialic acid signaling. All experiments were performed in the context of pancreatic cancer, as this malignancy is characterized by a large infiltrate of CAFs and an immune-suppressive tumor microenvironment (TME). Previously, these authors reported that pancreatic adenocarcinoma (PDAC) cells can facilitate immune suppression via similar sialic acid signaling in direct tumor-macrophage crosstalk. Here, the authors state that also CAFs in PDAC can affect anti-tumor immunity by interfering with monocyte-to-macrophage differentiation via sialic acid/siglec signaling. Although interesting, several major points need to be addressed to convince the readers of the proposed mechanism of action.

Major points:

1. For their studies, the authors make use of human pancreatic CAF cell lines (M1 and T1) and a normal pancreatic stellate cell line (PS-1). All cell lines were validated for their expression of vimentin, SMA, CD90, FAP and PDPN. The authors draw the conclusion that these cell lines represent CAFs in PDAC based on similar RNA expression profiles of aforementioned markers on the scRNAseq data of human PDAC of Peng et al. This statement is unsubstantiated as the scRNAseq dataset clearly shows heterogeneous expression of FAP and PDPN in the fibroblast cluster, which does not correlate with the expression of these markers in the individual cell lines. In fact, the whole heterogeneity of CAFs aspect is highly unstated thorough the manuscript. Especially considering the vast body of studies that ascribe immune-regulatory, myfibroblastic and anti-tumorigenic functions to PDAC CAFs (Hutton et al. Cancer Cell 2021, Elyada et al. Cancer Discovery 2019, Chen et al. Cancer Cell, 2021 and Biffi et al. Cancer Discovery 2019, amongst others). The authors should perform a better phenotyping of the cell lines that are so crucial to their study. Do these CAF lines represent iCAFs, myCAFs or apCAFs or a mix? And what about the PS-1 line? Do these represent normal stellate cells based on marker expression by comparing flow cytometry and scRNAseq data? And is this phenotype maintained throughout culture? And are the authors sure that the CAF lines are indeed CAFs and not EMT tumor cells (lack of cancer driver gene expression: P53 and KRAS mutation status)? If these analyses have been down in the past by the lab donating these cells, a reference to this should be included.
2. The authors state that, based on scRNAseq data, CAFs show the strongest expression of ST3GAL4 (figure 1F and supplemental figure 1D). Next they proceed to stain for ST3GAL4 on PDAC tissues where they observe ST3GAL4 expression in a subset of SMA+ CAFs only (figure 1G). Is this also evident from the scRNAseq dataset? Can a reclustering of the fibroblast cluster reveal CAF subsets and will this provide insight into which CAF subtype expresses ST3GAL4 (iCAFs vs myCAFs)? It is tempting to hypothesize that the iCAFs are the ST3GAL4 expressing cells considering their proposed immune-regulatory mechanisms. Furthermore, what are the levels of sialic acid-related genes in normal fibroblasts/stellate cells? Figure 1 A also indicated pan-Lectenz expression in adjacent normal tissue. Is there an upregulation of the sialic-acid pathway in CAFs compared to normal fibroblasts? Or is this

sialic-acid gene signature also present in normal fibroblasts with immune-related functions (for instance the Pi16+ pan-tissue fibroblasts described by Buechler et al (Nature 2021)).

3. Despite using two PDAC CAF cell lines for their studies, the authors observe the most convincing phenotype with the normal pancreatic stellate cell line PS-1, making their entire study based on 1 cell model. However, they continue to make the claim that this sialic acid/siglec signaling is relevant in CAFs, without showing this in an appropriate model system. The data would have been more convincing with multiple cell lines or human primary CAF cultures.

4. Section 'fibroblasts differentiate monocytes to suppressive macrophages resembling TAMs': the co-culture data is presented as a tSNE-plot in figure 3 and based on this clustering the authors state that the fact that monocytes co-cultured with PDAC cells or fibroblasts cluster together in the tSNE indicate that fibroblasts can also differentiate monocytes to TAMs. Perhaps I do not fully understand the data, but should one not expect co-cultured monocytes to cluster together with cytokine-induced macrophages if indeed fibroblasts are able to induce monocyte-to-macrophage differentiation?

5. Section 'fibroblasts differentiate monocytes to suppressive macrophages resembling TAMs': the authors continue to analyze Siglec expression on the TAMs of their co-cultures (figure 3D-G) and conclude that all macrophages from each co-culture express Siglec7 and 9. If Siglec expression on all TAMs of the co-cultures are comparable (PS-1 and CAF lines), then how much does this pathway really contribute to the observed phenotype of CAFs being able to induce macrophage differentiation? Especially considering that the secretome of both PS-1 and CAF cell lines are comparable (supplemental figure 3D) and all fibroblast cell lines express the sialic acids capable of activating Silec7, 9, 10 and 15 (figure 2D). What explains the differences in macrophage differentiation and IL10 secretion observed in figure 3D and E? Based on this data I would conclude that the sialic acid/siglec pathways cannot explain the observed differences and additional pathways are at play or am I missing something?

6. In their previous work the authors showed direct tumor cell-TAM crosstalk based on sialic acid/Siglec signaling leading to an immune-suppressive TME. How relevant are the CAFs in this? What is the relative contribution of tumor cells or CAFs to this immune-suppressive crosstalk with macrophages? Is the tumor cell-TAM crosstalk more dominant than the CAF-TAM crosstalk or do they function via different pathways? Can co-cultures of tumor cells, CAFs and monocytes together give insight into this? Especially if co-cultures were performed with sialic acid-related gene knockout tumor cells co-cultured with wildtype CAFs and monocytes versus wildtype tumor cells co-cultured with sialic-acid-related gene knockout CAFs and monocytes.

7. In the experiments performed with the sialyltransferase inhibitors, the most pronounced effect was seen with the T1 CAFs, even after the 3 day wash-out of the drug. However, inhibition of sialic acid signaling in these cells does not affect their ability to induce macrophage differentiation, thereby contradicting their hypothesis and conclusion of the study. Additional CAF cell lines or primary cultures should be include to convince the reader of the proposed mechanism of action and relevance of these CAFs to induce pro-tumorigenic macrophage differentiation.

8. The authors assess monocyte-to-macrophage differentiation by flow cytometry based on three markers (CD163, CD206 and CD68). To make a stronger case regarding the pro-tumorigenic macrophage differentiation induced by fibroblasts, the authors should also investigate macrophage functionality (like antigen-presentation, cytokine production or NO production) or at least show gene expression profiles and pathway analysis of the macrophages to confirm a pro-tumorigenic/M2-like phenotype.

9. Section 'abrogation of sialic acids on fibroblasts results in decreased differentiation to TAMs': Here the fibroblast cell lines are treated with sialyltransferase inhibitors. Please show that these inhibitors do not interfere with cell viability of the fibroblasts, because that could skew the outcome of the

experiments. The same goes for the knockdown/knockout of ST3GAL4 and Siglec9 (Figure 5).

10. Figure 4: for the SI inhibitor studies, the authors determine the effect of sialic acids re-appearing on the cells after washing out the drug for 3 days (schematics in figure 4A), but in the actual experiment where the monocytes are added (schematic in panel D), the readout is done after 4 days. Is this correct? Or a typo in the figure? If correct, than the effect of washing out the drug should also be assessed after 4 days, to be consisted amongst the experiments that are being compared.

Minor points:

1. Line 28 (abstract): remove word 'and' from sentence.
2. Line 66 (introduction): the authors state: 'these few studies highlight a role of CAFs in immune suppression...'. Please remove word 'few'.
3. Line 106 (material and methods): provide more information regarding the hT1 and hM1 CAF cell lines obtained from Prof. Tuveson. Either add reference or explain how these cell lines are generated/immortalized etc.
4. Figure 3A: it is not clear from the image or the text whether co-cultures were performed with only the indicated cell types together with monocytes or also co-cultures of for instance tumor cells, fibroblasts AND monocytes.
5. Line 265 (Results): Expression of ST3GAL4 was examined in dataset of Buechler et al. Please provide more info on which datasets were used (normal, tumor, both?), also in the material and methods, because this paper contains various datasets.
6. Line 452 (Discussion): add reference of Elyada et al. (Cancer Discovery 2019) describing iCAFs, myCAFs and apCAFs.
7. Supplemental figure 1E: please plot data as cluster averages, because this makes it more easy for readers to assess expression of target genes in rare clusters like the CCL19 cluster.
8. Supplemental figure 2E: please show this data for all cell lines used in the manuscript. Representing this type of flow cytometry data as MFI only (as done in main figure 2B) lacks information regarding positive and negative populations.
9. Throughout manuscript: the authors use the terms M1 and M2 to refer to pro- and anti-tumorigenic macrophages/TAMs. Multiple studies have shown that the image of M1 and M2 macrophages is not black-and-white and that various states occur in vivo. Therefore I believe the authors should use the terms pro- and anti-tumorigenic TAMs rather than M1 and M2. Furthermore, one of their CAF cell lines is also called M1, adding to the confusion.

Reviewer #2 (Remarks to the Author):

The authors describe the role of sialic acid on cancer-associated fibroblasts (CAFs). They describe an increased presence of sialic acid in the stromal compartment of pancreatic adenocarcinoma by lectin staining. They further studied genes associated with sialylation in datasets and found a correlation with CAFs and sialylation-associated genes. They further show Siglec-Fc binding to CAF cell lines and Siglec receptor expression on myeloid cells in cancer and monocyte derived primary cells. The authors further show that sialylation on CAF cell lines is associated with TAM differentiation by in vitro co-culture and either reducing sialic acid by sialidase and genetically or downregulating Siglec-9 in monocytes led to a reduced polarization toward TAMs. Although this an interesting manuscript, there are several questions that should be addressed before publication.

1. The main question is if sialic acid on CAFs is different/more relevant than on other cell types/ECM in the tumor microenvironment. Along with this general question, it would be important to understand the following points:
 - Are the ligands different on CAFs than on tumor cells?
 - Could e.g. in the experiment shown in Figure 5 C-E also tumor cells be used as tumor cells also bind to Siglec-9?
2. What happens when Siglec receptors are blocked with antibodies, e.g. Siglec-15 blocking antibodies (currently in clinical development for cancer)?
3. Which sialic acid-containing glycoconjugates are most important on CAFs, are these N-linked or O-linked glycoproteins?
4. How specific is CAF sialylation to only pancreatic cancer or could this also play a role in cancers?

Minor:

5. Abstract: I would not call a tumor microenvironment 'aggressive'. Rather immunosuppressive or mediating aggressiveness of the cancer.
6. In several analyses, a paired T test was used although multiple conditions were tested (e.g. 3D-F). In order to not only obtain statistical relevance for only a comparison to 1 condition, test for multiple testing could be used instead.

Reviewer #3 (Remarks to the Author):

In a previous study, the authors showed that overexpression of sialic acids in PDAC cells contributes to an immune suppressive microenvironment by promoting TAM differentiation via the interaction with Siglec-7 and Siglec-9.

In this manuscript the authors study the sialylation of CAFs and its role in immune modulation of myeloid cells in PDAC.

The authors show that PDAC cell lines of CAFs and pancreatic stellated cells (PSC) abundantly express sialic acid-containing glycans, which stimulates the differentiation of monocytes to M2-like TAMs contributing to immune suppression via binding to Siglec receptors.

Several points should be addressed in order to improve the manuscript.

1. In figure 1 and supplemental figure 1 the number of patients analyzed should be stated. The stage of the lesions should be stated. Furthermore the quantification of the IHC staining in the stroma and in the tumoral compartment should be done in order to understand what is the proportion of tumor and stroma expressing sialic acid.
In figure 1 there is an important staining in the adjacent tissue as well. What is the significance of this staining? Can the authors comment on that?
2. The author use public data. There is no mention of how many patients are included. In general the Transcriptomic data analysis is very vague and need further development of the algorithms used.

3. The authors state that ST3GAL4 was expressed in several α -SMA+ cells in the PDAC tissues. Quantification of the % of ST3GAL4+ in α -SMA cells should be performed in the 6 analyzed patients.

4. The authors use CAF and pancreatic stellate cells (PSC) cell lines. The authors show that all the cell line (CAFs and PSC) express α SMA. PSC are quiescent cells and should not express α -SMA. What about the expression of GFAP which is a marker of quiescent PSC?

5. In figure 2E the authors show representative multiplex immunohistochemistry. In order to state the expression of Siglec-7, 10, 9 and 15 on CD14 the quantification of the % of positivity among CD14 cells in all analyzed patients (what " \geq " means?).

6. The authors claim that abrogation of sialic acid on fibroblast leads to decreased M2 differentiation of macrophages (Fig 4 and Supplemental Fig.4)

However the abrogation of sialic acid on CAFs (T1 and M1) has no impact of M2 phenotype.

There is only an impact of the abrogation of sialic acid on PSC on M2 phenotype.

This suggest that the M2 phenotype induction by CAFs is not dependent on sialic acid expression.

As the CMAS KO was not performed in CAFs (due to technical problems) it is impossible to conclude on the role of sialic acid in the M2 phenotype. Furthermore, Siglec-9 KO in monocytes reduced fibroblast-mediated TAM differentiation in co-culture with PS-1 cells, but not in co-culture with M1 or T1 CAFs.

Therefore in the discussion the paragraph is an overinterpretation since this the data show that only in PSC presence the M2 phenotype of TAMs was dependent on sialic acids.

" Removing CAF sialylation by sialyltransferase inhibitor treatment or through genetic KO of key regulators such as ST3GAL4 and CMAS that alter the sialylation pathway, resulted in reduced differentiation of monocytes to M2-like TAMs. Our work thus identifies a novel CAF-immune crosstalk dependent on glycosylation of CAFs and the interaction with suppressive receptors on immune cells".

Authors should consider the differences between PSC and CAFs and the involvement of those cells in early stages (ADM acinar to ductal metaplasia) than in late stages (PDAC).

Furthermore, the following conclusion is an overstatement.

"Based on marker expression and cytokine profiles, there was no logical explanation for this distinction between PS-1, M1 and T1 CAFs". The activation status of PSC compared to CAF should be taken into account (gel contraction, GFAP expression, T cell inhibition etc..).

7. The study lack in vivo relevance of the observed phenotype.

Response to reviewers

Reviewer #1 (Remarks to the Author):

Review report

Manuscript title: pancreatic cancer-associated fibroblasts modulate macrophage differentiation via sialic acid-Siglec interactions

Manuscript ID: COMMSBIO-23-0459-T

Reviewer: Dr. Julia Houthuijzen, Molecular Pathology, The Netherlands Cancer Institute.

In this manuscript the authors identified immune-modulatory effects of cancer-associated fibroblasts (CAFs) on macrophages via sialic acid signaling. All experiments were performed in the context of pancreatic cancer, as this malignancy is characterized by a large infiltrate of CAFs and an immune-suppressive tumor microenvironment (TME). Previously, these authors reported that pancreatic adenocarcinoma (PDAC) cells can facilitate immune suppression via similar sialic acid signaling in direct tumor-macrophage crosstalk. Here, the authors state that also CAFs in PDAC can affect anti-tumor immunity by interfering with monocyte-to-macrophage differentiation via sialic acid/siglec signaling. Although interesting, several major points need to be addressed to convince the readers of the proposed mechanism of action.

Dear Dr. Julia Houthuijzen,

We sincerely appreciate the time and effort you invested in reviewing our manuscript. Your insightful comments have been crucial, and we are grateful for the thoroughness of your review. The revised manuscript now incorporates new data and analysis to answer your concerns, including additional analysis on sialylation in CAF subsets (Figure 1H-I, supplementary Figure 1F-H); better characterisation of CAF cell lines (Supplementary Figure 2C-F), their distinct glycosylation profiles (Figure 3A) and Siglec-9 ligands (Figure 3B); validation of the immunosuppressive function of CAF-induced TAMs (Figure 4H); and differences in Siglec ligands of CAF versus tumor (Figure 3B) and its relative contribution in TAM differentiation (Figure 6).

We noted some uncertainty regarding the use of PS-1 cells as model for CAFs in our manuscript. We have further clarified on this topic in this rebuttal and included additional data confirming the rationale of using PS-1 as a model for CAFs.

Furthermore, we noticed that our conclusion regarding the effect of sialic acids on CAF-driven TAM differentiation was not fully appreciated. In this revised manuscript, we have included new data that better characterizes the glycosylation profile of the CAF cell lines, revealing distinct Siglec-9 ligands between them (Figure 3A-B). Consequently, we included a more nuanced discussion on the results and the complexity of glycosylation-immune interactions. For clarity of the manuscript, we emphasize the role of sialylation in TAM differentiation in context of PS-1 CAFs, and relate it to tumor sialylation. We moved data with other CAF cell lines to supplementary Figure 5.

We are grateful for your input and we hope that our rebuttal adequately addresses your questions and concerns.

Sincerely,

Prof.dr. Yvette van Kooyk

Major points:

1. For their studies, the authors make use of human pancreatic CAF cell lines (M1 and T1) and a normal pancreatic stellate cell line (PS-1). All cell lines were validated for their expression of vimentin, SMA, CD90, FAP and PDPN. The authors draw the conclusion that these cell lines represent CAFs in PDAC based on similar RNA expression profiles of aforementioned markers on the scRNAseq data of human PDAC of Peng et al. This statement is unsubstantiated as the scRNAseq dataset clearly shows heterogeneous expression of FAP and PDPN in the fibroblast cluster, which does not correlate with the expression of these markers in the individual cell lines. In fact, the whole heterogeneity of CAFs aspect is highly unstated thorough the manuscript. Especially considering the vast body of studies that ascribe immune-regulatory, myofibroblastic and anti-tumorigenic functions to PDAC CAFs (Hutton et al. Cancer Cell 2021, Elyada et al. Cancer Discovery 2019, Chen et al. Cancer Cell, 2021 and Biffi et al. Cancer Discovery 2019, amongst others). The authors should perform a better phenotyping of the cell lines that are so crucial to their study. Do these CAF lines represent iCAFs, myCAFs or apCAFs or a mix? And what about the PS-1 line? Do these represent normal stellate cells based on marker expression by comparing flow cytometry and scRNAseq data? And is this phenotype maintained throughout culture? And are the authors sure that the CAF lines are indeed CAFs and not EMT tumor cells (lack of cancer driver gene expression: P53 and KRAS mutation status)? If these analyses have been down in the past by the lab donating these cells, a reference to this should be included.

We appreciate the reviewer's comments. In response, we have substantially expanded the characterisation of our cell lines (supplementary Figure 2C-F, Figure 3A-B), showing distinct phenotypes, activation states [line 344-375 in result section], and glycosylation profiles (Figure 3A-B). We included the reference describing immortalization of the M1 and T1 CAFs, which provides evidence of their lack of cancer driver mutations and absence of tumor outgrowth in vivo [line 122-124].

Pancreatic stellate cells are known to acquire an activated myofibroblast phenotype when cultured as a monolayer, and therefore can be used as a model for myCAFs (1, 2). We have included additional phenotyping of the PS-1, that confirm the loss of a quiescent phenotype (as indicated by the lack GFAP, supplementary Figure 2C), and the acquirement of an activated myofibroblastic phenotype (evaluated using gel contraction assay, supplementary Figure 2D). These additional characterizations, coupled with the expression of established CAF markers such as α -SMA, FAP, and CD90 (Supplementary Figure 2E-F), collectively validate the rationale for using PS-1 cells as a suitable model for CAFs in our study, and we clarified the text by referring to this cell line as PS-1 CAFs.

We also further characterized the phenotype of M1 and T1 CAF cell lines (supplementary Figure 2C-F). The CAF cell lines are not apCAFs, given the lack of HLA-DR expression (supplementary Figure 2H). However, it is challenging to assigning them as iCAFs or myCAFs, given that they express markers associated with both phenotypes (supplementary Figure 2B,E-G). Therefore, we concluded that the 3 cell lines show distinct characteristics and activation, expressing both myCAFs and iCAFs markers, with the PS-1 CAF having the most activated phenotype of the three cell lines.

The most important observation regarding the CAF cell lines in the context of our study, was the distinct glycosylation profiles of the cell lines (Figure 3A), and the differences in Siglec-9 ligands (Figure 3B), further discussed below.

We acknowledge that cells lines will not recapitulate the full heterogeneity observed in vivo, and that this is a limitation of our study, which we highlighted in the discussion [line 612-624]. We now also describe CAF subsets in the introduction of the revised manuscript and included additional literature references [line 56-60]

2. The authors state that, based on scRNAseq data, CAFs show the strongest expression of ST3GAL4 (Figure 1F and supplemental Figure 1D). Next they proceed to stain for ST3GAL4 on PDAC tissues where they observe ST3GAL4 expression in a subset of SMA+ CAFs only (Figure 1G). Is this also evident from the scRNAseq dataset? Can a reclustering of the fibroblast cluster reveal CAF subsets and will this provide insight into which CAF subtype expresses ST3GAL4 (iCAFs vs myCAFs)? It is tempting to hypothesize that the iCAFs are the ST3GAL4 expressing cells considering their proposed immune-regulatory mechanisms. Furthermore, what are the levels of sialic acid-related genes in normal fibroblasts/stellate cells? Figure 1A also indicated pan-Lectenz expression in adjacent normal tissue. Is there an upregulation of the sialic-acid pathway in CAFs compared to normal fibroblasts? Or is this sialic-acid gene signature also present in normal fibroblasts with immune-related functions (for instance the Pi16+ pan-tissue fibroblasts described by Buechler et al (Nature 2021)).

We have performed additional analysis on the scRNA-seq dataset, which show that both myCAFs and iCAFs exhibit enhanced α 2-3 sialylation compared to normal CAFs (Figure 1H), with no significant differences observed between CAF subsets (supplementary Figure 1H). Furthermore, ST3GAL4 expression was increased in CAFs compared to NAFs (Figure 1I), and interestingly, myCAFs showed significantly higher expression levels than iCAFs (supplementary Figure 1H).

These results suggest an increased sialylation in CAFs, with subtle differences among subsets. Consequently, we have refrained from placing substantial emphasis on CAF subsets in the revised manuscript. We did not observe a distinct apCAF cluster in this dataset (supplementary Figure 1F), consistent with findings from Elyada et al. 2019 in their human PDAC scRNA-seq dataset analysis.

3. Despite using two PDAC CAF cell lines for their studies, the authors observe the most convincing phenotype with the normal pancreatic stellate cell line PS-1, making their entire study based on 1 cell model. However, they continue to make the claim that this sialic acid/siglec signaling is relevant in CAFs, without showing this in an appropriate model system. The data would have been more convincing with multiple cell lines or human primary CAF cultures.

Having established above that PS-1 represent the most activated myofibroblastic cell line across the three cell lines, we are therefore convinced that the PS-1 CAFs are an appropriate model for CAFs. Furthermore, ST3GAL4 was significantly higher in myCAFs than iCAFs (supplementary Figure 1H), further suggesting PS-1 as the most representative model to study sialylation of CAFs given its myofibroblastic phenotype.

For the contrasting results between the three CAF cell lines, we hypothesized that differences in glycosylated ligands may explain these results, given the complexity in glycan-immune interactions. Siglec binding to its ligand, sialic acid, and signalling is dependent on various factors including linkage (α 2-3, α 2-6, α 2-8), underlying glycan and protein or lipid carrier (1) [included in intro line 92-94]. As observed in Figure 3A, the glycosylation profile was indeed different in PS-1 CAFs than M1/T1 CAFs, with less diversity in PS-1 CAFs. Moreover, we found that the specific ligand(s) for Siglec-9 were different in PS-1 compared to M1/T1 CAFs (Figure 3B), with them being mainly in N-glycans in the former but distributed among different glycan structures in the latter. These differences can explain the observed differences in biological consequences. For this reason and based on your suggestion, we now highlight better that PS-1 is more clear system to prove biological effects of sialylation (see also comment 7), based on having N-glycosylation as ligand for Siglec-9. We have modified the Figures to focus on PS-1, and included the data with M1/T1 CAFs supplementary.

4. Section 'fibroblasts differentiate monocytes to suppressive macrophages resembling TAMs': the co-culture data is presented as a tSNE-plot in Figure 3 and based on this clustering the authors state that the fact that monocytes co-cultured with PDAC cells or fibroblasts cluster together in the tSNE

indicate that fibroblasts can also differentiate monocytes to TAMs. Perhaps I do not fully understand the data, but should one not expect co-cultured monocytes to cluster together with cytokine-induced macrophages if indeed fibroblasts are able to induce monocyte-to-macrophage differentiation?

We appreciate the reviewer's careful consideration of the tSNE-plot and the associated interpretation. Given the relative high cytokine concentrations that are used in the differentiation of monocytes, we would not expect them to cluster together with monocytes derived from the co-culture with fibroblast cell lines, and they are mainly used as positive controls and a reference during in the analysis. However, in this context, our data show that the fibroblast-induced macrophages present an intermediate phenotype between monocytes and M-CSF-induced macrophages. Their clustering away from M1- and M2- moMACs is mainly due to upregulation of PD-L1 by the cytokines, which makes them cluster distinct from the rest (2).

We chose not to include the illustration in the manuscript due to potential misinterpretation arising from color-scaling influences on tSNE plots. Adjusting the scaling highlighted the difference in intensity, but it risked creating a misleading impression that co-cultured monocytes lack PD-L1 expression.

5. Section 'fibroblasts differentiate monocytes to suppressive macrophages resembling TAMs': the authors continue to analyze Siglec expression on the TAMs of their co-cultures (Figure 3D-G) and conclude that all macrophages from each co-culture express Siglec7 and 9. If Siglec expression on all TAMs of the co-cultures are comparable (PS-1 and CAF lines), then how much does this pathway really contribute to the observed phenotype of CAFs being able to induce macrophage differentiation? Especially considering that the secretome of both PS-1 and CAF cell lines are comparable (supplemental Figure 3D) and all fibroblast cell lines express the sialic acids capable of activating Siglec7, 9, 10 and 15 (Figure 2D). What explains the differences in macrophage differentiation and IL10 secretion observed in Figure 3D and E? Based on this data I would conclude that the sialic acid/siglec pathways cannot explain the observed differences and additional pathways are at play or am I missing something?

We appreciate the reviewer's question. The aim of Figure 4 (previously Figure 3) is to show that CAFs are able to induce the differentiation of CD163⁺CD206⁺ macrophages and IL-10 secretion, and not necessarily the involvement of sialic acid (which is studied in later Figures). CAF-mediated TAM differentiation is a multifactorial process involving various pathways and interactions, and our manuscript reports that sialic acids-Siglec interactions are contributing in this process, as has been also described in literature (3-5).

Furthermore, there is an underlying complexity in sialic acid-Siglec interactions, in which subtle differences can affect the biological role of these glycans. For example, Siglec binding and signalling is not only dependent on sialic acid presence, but also on its underlying glycan and carrier (as further

discussed in comment 7). Therefore, the presence of Sialic acid (as measured using lectins or by glycomics) or Siglec ligands (using Fc chimeras), cannot be directly translated to the activation of Siglecs. Moreover, the three CAF cell lines have different activation states and phenotypes which can also underlie the small differences in CAF-induced TAM phenotypes observed in Figure 4D-E.

6. In their previous work the authors showed direct tumor cell-TAM crosstalk based on sialic acid/Siglec signaling leading to an immune-suppressive TME. How relevant are the CAFs in this? What is the relative contribution of tumor cells or CAFs to this immune-suppressive crosstalk with macrophages? Is the tumor cell-TAM crosstalk more dominant than the CAF-TAM crosstalk or do they function via different pathways? Can co-cultures of tumor cells, CAFs and monocytes together give insight into this? Especially if co-cultures were performed with sialic acid-related gene knockout tumor cells co-cultured with wildtype CAFs and monocytes versus wildtype tumor cells co-cultured with sialic-acid-related gene knockout CAFs and monocytes.

We appreciate the reviewer's insightful question. To address the relative contribution of CAF sialylation versus tumor sialylation, we performed additional experiments which indicate that both tumor and CAF sialylation contribute to TAM differentiation (Figure 6B,D), and suggest that CAF sialylation plays a more crucial role in influencing TAM differentiation compared to tumor cells. Notably, the Siglec-9 ligands on tumor cells are different than those found on CAFs (Figure 3B).

Moreover, considering the distinct spatial distribution within the tumor microenvironment, we highlight in Figure 2F that the majority of myeloid cells are located in the stromal compartment rather than within tumor islands. This spatial distinction emphasizes the impact of stromal sialylation on myeloid cell phenotype, especially considering that in PDAC, the stroma can constitute up to 80% of the tumor mass.

7. In the experiments performed with the sialyltransferase inhibitors, the most pronounced effect was seen with the T1 CAFs, even after the 3 day wash-out of the drug. However, inhibition of sialic acid signaling in these cells does not affect their ability to induce macrophage differentiation, thereby contradicting their hypothesis and conclusion of the study. Additional CAF cell lines or primary cultures should be include to convince the reader of the proposed mechanism of action and relevance of these CAFs to induce pro-tumorigenic macrophage differentiation.

We hypothesized that differences in ligands may underlie the contrasting results, given the complexity in glycan-immune interactions. Siglec binding to its ligand, sialic acid, is dependent on various factors including linkage (α 2-3, α 2-6, α 2-8), underlying glycan and protein or lipid carrier (1).

Further characterisation of the glycosylation profile and Siglec-ligands on the three CAF cell lines revealed more diversity in glycosylation patterns in M1/T1 CAFs compared to PS-1 CAFs (Figure 3A), as well as distinct Siglec-9 ligands (Figure 3B). In the process of glycosylation, proteins can be glycosylated in an N-linked manner (N-glycosylation), or in an O-linked manner (O-glycosylation). In case of lipid glycosylation these glycans are called glycosphingolipids. Interestingly, in the PS-1 CAFs, N-glycosylated proteins were the major ligand of Siglec-9, while this was not the case in M1 and T1 CAFs (Figure 3B). N-glycosylation also showed to be particularly enriched in CAFs based on transcriptomic analysis (Figure 3C). The abundantly expressed glycans showed to be present in all the CAF cell lines (Figure 3A), suggesting a prominent role for differences in glycoprotein ligands in the interaction with Siglec-9 in PS-1 cells compared to M1/T1 CAFs. Supporting this idea, we observed differential expression of several highly glycosylated proteins (PDPN, CD106, CD146) between CAF cell lines (supplementary Figure 2F). Identification of the glycoprotein ligands involves a study on its own as there are many candidates. We have incorporated this in the discussion [line 601-611].

Together, these data underscore the relevance of N-glycosylation of CAFs. As such, we particularly emphasize in the role of sialic acids on PS-1 CAFs as these cells show prominent involvement of N-glycans in Siglec-9 binding, and we have moved the M1 and T1 CAF data to supplementary Figure 5.

8. The authors assess monocyte-to-macrophage differentiation by flow cytometry based on three markers (CD163, CD206 and CD68). To make a stronger case regarding the pro-tumorigenic macrophage differentiation induced by fibroblasts, the authors should also investigate macrophage functionality (like antigen-presentation, cytokine production or NO production) or at least show gene expression profiles and pathway analysis of the macrophages to confirm a pro-tumorigenic/M2-like phenotype.

We thank the reviewer for this comment. To address the functionality of CAF-induced TAMs, we have conducted new experiments evaluating the effect of the macrophages on CD8-T cell proliferation. This new data (Figure 4H) demonstrates that the TAMs differentiated with fibroblast conditioned media show a functional phenotype resembling that of M2-moMACs (pro-tumorigenic macrophages), highlighting their immunosuppressive phenotype.

We have taken into account the other suggestions of the reviewer, including cytokine and NO production. IL-12 and TNF α were measured using ELISA but were not detected (data not shown). As the CAFs also produce IL-6 and IL-8, it was more challenging to measure IL-6 and IL-8 production by the macrophages in the co-culture. Human monocyte-derived macrophages do not produce NO (6).

9. Section 'abrogation of sialic acids on fibroblasts results in decreased differentiation to TAMs': Here the fibroblast cell lines are treated with sialyltransferase inhibitors. Please show that these inhibitors do not interfere with cell viability of the fibroblasts, because that could skew the outcome of the experiments. The same goes for the knockdown/knockout of ST3GAL4 and Siglec9 (Figure 5).

We acknowledge the importance of ensuring that our experimental treatments do not impact cell viability, which could potentially introduce bias into our results. In response to this concern, we have incorporated supplementary data (supplementary Figure 5A) demonstrating that the sialyltransferase inhibitor used in our experiments does not affect CAF viability, granularity, size, or morphology. Additionally, we have included cell viability data for both ST3GAL4 siRNA knockdown and Siglec-9 knockout (supplementary Figure 5H, J).

10. Figure 4: for the SI inhibitor studies, the authors determine the effect of sialic acids re-appearing on the cells after washing out the drug for 3 days (schematics in Figure 4A), but in the actual experiment where the monocytes are added (schematic in panel D), the readout is done after 4 days. Is this correct? Or a typo in the Figure? If correct, than the effect of washing out the drug should also be assessed after 4 days, to be consisted amongst the experiments that are being compared.

Indeed, we assessed sialic acid re-appearance at the 3-day time point, to provide insights into the kinetics of sialic acid re-appearance during the co-culture period. Measuring sialic acid re-appearance at the end of the 4-day assay might reflect synthesis occurring in the final hours, having minimal impact on macrophage differentiation.

Minor points:

- 1. Line 28 (abstract):** remove word 'and' from sentence.

We have removed the word 'and'.

- 2. Line 66 (introduction):** the authors state: 'these few studies highlight a role of CAFs in immune suppression...'. Please remove word 'few'.

We have removed the word 'few'.

- 3. Line 106 (material and methods):** provide more information regarding the hT1 and hM1 CAF cell lines obtained from Prof. Tuveson. Either add reference or explain how these cell lines are generated/immortalized etc.

We have included the reference that describes these cell lines.

- 4. Figure 3A:** it is not clear from the image or the text whether co-cultures were performed with only the indicated cell types together with monocytes or also co-cultures of for instance tumor cells, fibroblasts AND monocytes.

We have modified the image and Figure legends to enhance the clarity of the experimental setup.

- 5. Line 265 (Results):** Expression of ST3GAL4 was examined in dataset of Buechler et al. Please provide more info on which datasets were used (normal, tumor, both?), also in the material and methods, because this paper contains various datasets.

Given that this dataset only included 3 PDAC patients, we have removed this data from the manuscript.

- 6. Line 452 (Discussion):** add reference of Elyada et al. (Cancer Discovery 2019) describing iCAFs, myCAFs and apCAFs.

We have included the reference.

- 7. Supplemental Figure 1E:** please plot data as cluster averages, because this makes it more easy for readers to assess expression of target genes in rare clusters like the CCL19 cluster.

Given that this dataset only included 3 PDAC patients, we have removed this data from the manuscript.

- 8. Supplemental Figure 2E:** please show this data for all cell lines used in the manuscript. Representing this type of flow cytometry data as MFI only (as done in main Figure 2B) lacks information regarding positive and negative populations.

We have included the histograms for all the cell lines in supplementary Figure 2J.

- 9. Throughout manuscript:** the authors use the terms M1 and M2 to refer to pro- and anti-tumorigenic macrophages/TAMs. Multiple studies have shown that the image of M1 and M2 macrophages is not black-and-white and that various states occur in vivo. Therefore I believe the

authors should use the terms pro- and anti-tumorigenic TAMs rather than M1 and M2. Furthermore, one of their CAF cell lines is also called M1, adding to the confusion.

We thank the reviewer for this suggestion and have adjusted the terminology used throughout the manuscript.

## Reviewer #2 (Remarks to the Author):

**The authors describe the role of sialic acid on cancer-associated fibroblasts (CAFs). They describe an increased presence of sialic acid in the stromal compartement of pancreatic adenocarcinoma by lectin staining. They further studied genes associated with sialylation in datasets and found a correlation with CAFs and sialylation-associated genes. They further show Siglec-Fc binding to CAF cell lines and Siglec receptor expression on myeloid cells in cancer and monocyte derived primary cells. The authors further show that sialylation on CAF cell lines is associated with TAM differentiation by in vitro co-culture and either reducing sialic acid by sialidase and genetically or downregulating Siglec-9 in monocytes led to a reduced polarization toward TAMs. Although this an interesting manuscript, there are several questions that should be addressed before publication.**

*Dear reviewer #2,*

*We thank you for your time and effort you invested in reviewing our manuscript. Your insightful questions have significantly contributed to the improvement of our work. The revised manuscript now includes two additional Figures addressing the questions you raised, including characterisation glycosylation profile of CAF cell lines using mass spectrometry (Figure 3A); Siglec-9 ligands on O-linked, N-linked or glycolipids on tumor cells versus CAFs (Figure 3B), glycosylation pathway score in tumor versus CAFs (Figure 3C), and relative contribution of CAF versus tumor sialylation in TAM differentiation (Figure 6).*

*We believe these revisions substantially strengthen the overall quality of the manuscript. We look forward to your feedback on the revised manuscript and appreciate your continued engagement with our research.*

*Sincerely,*
*Prof.dr. Yvette van Kooyk*

**1. The main question is if sialic acid on CAFs is different/more relevant than on other cell types/ECM in the tumor microenvironment. Along with this general question, it would be important to understand the following points:**
**- Are the ligands different on CAFs than on tumor cells?**
**- Could e.g. in the experiment shown in Figure 5 C-E also tumor cells be used as tumor cells also bind to Siglec-9?**

*We appreciate the reviewer's insightful questions. In the revised manuscript, we have included a glycomics analysis of the CAF cell lines (Figure 3A), and a better characterisation of Siglec-9 ligands on CAFs and tumor cells using specific inhibitors (Figure 3B). We found that Siglec-9 ligands on PS-1 CAFs are predominantly present in N-glycan structures, while they are presented in the context of O-glycans in the BxPC3 tumor cells. These results show that the specific glycans that serve as Siglec-9 ligands are different in CAF and tumor cells.*

*Besides the glycan structure, the protein or lipid carriers may vary between tumor cells and CAFs. Tumor cells commonly express highly O-glycosylated mucins, which serve as key ligands for Siglecs, including Siglec-9 (7). In contrast, CAFs exhibit limited mucin*

expression (see plot on the right) and present the Siglec ligands in other carriers, which may change the presentation and clustering of these glycan structures.

Several O-glycan structures are similar on CAFs and tumor cells (Figure 3A, (8)), and also Siglec-7 ligands were presented on O-glycans for both (Supplementary Figure 3B, (8)).

To address the relative contribution of stromal sialylation versus tumor sialylation, we performed additional experiments, which indicate that both tumor and CAF sialylation contribute to TAM differentiation (Figure 6B,D), and suggest that while tumor and CAF sialylation are essential for contact-dependent TAM differentiation, CAF-derived sialylation plays a more dominant role in TAM differentiation through secreted products.

Moreover, considering the distinct spatial distribution within the tumor microenvironment, we highlight in Figure 2F that the majority of myeloid cells are located in the stromal compartment rather than within tumor islands. This spatial distinction emphasizes the impact of stromal sialylation on myeloid cell phenotype, especially considering that in PDAC, the stroma can constitute up to 80% of the tumor mass.

2. What happens when Siglec receptors are blocked with antibodies, e.g. Siglec-15 blocking antibodies (currently in clinical development for cancer)?

This is an intriguing question. Unfortunately, there are currently no commercially available blocking antibodies for Siglec-15 and Siglec-10. Despite our attempts to block Siglec-7 and Siglec-9 using available antibodies, their reliability is questionable, as agonistic properties have been reported, especially for the Siglec-9 antibody (9).

We performed the experiments with blocking antibodies Siglec-7 (Biolegend, cat# 347702) or Siglec-9 (R&D Systems, cat# MAB1139-100), with a final concentration of 2.5 $\mu\text{g}/\text{mL}$:

Interestingly, we observed a reduction in TAM differentiation with Siglec-9 blockade in the PS-1 cell line, but not with Siglec-7 blockade, suggesting that Siglec-9 may play a more prominent role in CAF-mediated TAM differentiation than Siglec-7. Given the uncertainties regarding the blocking capacity of these antibodies, we have not included this data in the revised manuscript.

Another essential consideration is that Siglec-15 is not expressed on monocytes (supplementary Figure 3A). We therefore speculate that Siglec-15 could play a role in macrophage polarization at later stages, rather than in the early steps of macrophage differentiation, as mentioned in the discussion [line 585-588].

3. Which sialic acid-containing glycoconjugates are most important on CAFs, are these N-linked or O-linked glycoproteins?

We thank the reviewer for this question. In response, we have performed additional experiments which show that N-glycosylated proteins were the major ligand of Siglec-9 in the PS-1 CAFs (Figure 3B). The M1 and T1 CAFs showed redundancy in Siglec-9 ligands. Transcriptional data analysis of glycosylation pathways demonstrated enriched N-glycosylation and glycolipid synthesis in fibroblast compared to tumor cells (Figure 3C), suggestion that N-linked glycoproteins and glycolipids are most important sialic acid-containing glycoconjugates in CAFs.

4. How specific is CAF sialylation to only pancreatic cancer or could this also play a role in cancers?

Recent studies indicate that sialylation of the stroma can also play a role in other cancer types. We have expanded our discussion to include these findings, suggesting a potential role for CAF sialylation in multiple cancer types [line 635-637].

Minor:

5. Abstract: I would not call a tumor microenvironment 'aggressive'. Rather immunosuppressive or mediating aggressiveness of the cancer.

We have revised the abstract, and changed the word aggressive to immunosuppressive.

6. In several analyses, a paired T test was used although multiple conditions were tested (e.g. 3D-F). In order to not only obtain statistical relevance for only a comparison to 1 condition, test for multiple testing could be used instead.

We appreciate the reviewer's attention to statistical analysis. We have adjusted the statistical tests used for Figure 4E-F (previously Figure 3E-F) to one-way repeated measures ANOVA and Dunnett's multiple comparison test. In Figure 4D a paired T test was used, given that repeated-measures ANOVA cannot handle missing values. In this case, we adjusted the threshold for significance using the Bonferroni-corrected p-value.

Reviewer #3 (Remarks to the Author):

In a previous study, the authors showed that overexpression of sialic acids in PDAC cells contributes to an immune suppressive microenvironment by promoting TAM differentiation via the interaction with Siglec-7 and Siglec-9.

In this manuscript the authors study the sialylation of CAFs and its role in immune modulation of myeloid cells in PDAC.

The authors show that PDAC cell lines of CAFs and pancreatic stellated cells (PSC) abundantly express sialic acid-containing glycans, which stimulates the differentiation of monocytes to M2-like TAMs contributing to immune suppression via binding to Siglec receptors.

Dear Reviewer #3,

We appreciate your thoughtful assessment of our manuscript. We want to express our gratitude for your time and consideration.

The revised manuscript now incorporates new data and analysis to answer the concerns, including better characterisation of CAF cell lines (Supplementary Figure 2C-F), their glycosylation profile (Figure 3A) and Siglec-9 ligands (Figure 3B); relative contribution of CAF versus tumor sialylation in TAM differentiation (Figure 6); and quantifications of multiplex immunofluorescent stainings (Figure 1B, 1J, 2F-G).

We noted some uncertainty with the reviewers regarding the use of PS-1 cells as model for CAFs in our manuscript. We have further clarified on this topic in this rebuttal and included additional data confirming the rational of using PS-1 as a model for CAFs.

Furthermore, we noticed that our conclusion regarding the effect of sialic acids on CAF-driven TAM differentiation was not fully appreciated. In response, we have further investigated the glycosylation profile, revealing distinct Siglec-9 ligands between CAF cell lines (Figure 3A-B). Consequently, we included a more nuanced discussion on the results and the complexity of glycosylation-immune interactions. For clarity of the manuscript, we emphasize the role of sialylation in TAM differentiation in context of PS-1 CAFs, and relate it to tumor sialylation. We moved data with other CAF cell lines to supplementary Figure 5.

We look forward to your feedback on the revised version and appreciate your continued engagement with our work.

*Sincerely,
Prof.dr. Yvette van Kooyk*

Several points should be addressed in order to improve the manuscript.

1. In Figure 1 and supplemental Figure 1 the number of patients analyzed should be stated. The stage of the lesions should be stated. Furthermore the quantification of the IHC staining in the stroma and in the tumoral compartment should be done in order to understand what is the proportion of tumor and stroma expressing sialic acid.

In Figure 1 there is an important staining in the adjacent tissue as well. What is the significance of this staining? Can the authors comment on that?

We thank the reviewer for this insightful comment. We have now included information on lesion stage (supplementary table 1) and quantified sialylation in both stromal and tumoral compartments (Figure 1B), showing higher sialic acid presence in stroma than in tumor cells, emphasizing the significance of stromal sialylation.

The staining in adjacent tissue reveals sialic acid levels also in normal-adjacent stroma, although the quantity of stroma is limited. We have included additional analysis of the scRNA-seq data showing significantly enriched sialylation-related gene expression in CAFs compared to normal fibroblast (Figure 1H-I). These data further highlight the enriched sialylation of CAFs in tumor tissue when compared with fibroblast in the normal counterpart.

2. The author use public data. There is no mention of how many patients are included. In general the Transcriptomic data analysis is very vague and need further development of the algorithms used.

We appreciate the reviewer's input. In response to the comment, we have included additional information in the Methods section [line 256-274], specifying the number of patients included and providing more details on the algorithms used for transcriptomic data analysis.

3. The authors state that ST3GAL4 was expressed in several α -SMA+ cells in the PDAC tissues. Quantification of the % of ST3GAL4+ in α -SMA cells should be performed in the 6 analyzed patients.

In response to your comment, we have now included quantification of ST3GAL4 expression in stromal cells of PDAC tissues from the six analyzed patients. Notably, quantifying fibroblasts in tissue stainings presents certain challenges due to the heterogeneous morphologies of these cells, making it difficult to confidently assign a single nucleus to the α -SMA signal. Therefore, we quantified ST3GAL4 expression in immune cells (CD45) and tumor cells (panCK). For the stromal cells not expressing CD45, we assigned them as stromal cells expressing ST3GAL4.

4. The authors use CAF and pancreatic stellate cells (PSC) cell lines. The authors show that all the cell line (CAF and PSC) express α SMA. PSC are quiescent cells and should not express α -SMA. What about the expression of GFAP which is a marker of quiescent PSC?

It is indeed acknowledged that PSCs are typically quiescent cells and do not express α -SMA in their native state. Notably, PSC acquire an activated myofibroblast phenotype when cultured in monolayer, and therefore can be used as a model for myCAF (10). We have included additional phenotyping of the cell lines, which indeed highlight that the PS-1 cells lost their quiescent phenotype (lack GFAP, manuscript supplementary Figure 2C), and acquired an activated myofibroblastic phenotype (gel contraction assay, supplementary Figure 2D).

These additional characterizations, coupled with the expression of established CAF markers such as α -SMA, FAP, and CD90 (Supplementary Figure 2E-F), collectively validate the transition of PS-1 cells to a myofibroblast state. We believe that these clarifications further strengthen the rationale for using PS-1 cells as a suitable model for CAFs in our study.

5. In Figure 2E the authors show representative multiplex immunohistochemistry. In order to state the expression of Siglec-7, 10, 9 and 15 on CD14 the quantification of the % of positivity among CD14 cells in all analyzed patients (what " \geq " means?).

In response to your valuable suggestion, we have incorporated the quantification of Siglec expression among CD14 cells in the revised manuscript (Figure 2G).

6. The authors claim that abrogation of sialic acid on fibroblast leads to decreased M2 differentiation of macrophages (Fig 4 and Supplemental Fig.4)

However the abrogation of sialic acid on CAFs (T1 and M1) has no impact of M2 phenotype. There is only an impact of the abrogation of sialic acid on PSC on M2 phenotype. This suggest that the M2 phenotype induction by CAFs is not dependent on sialic acid expression.

As the CMAS KO was not performed in CAFs (due to technical problems) it is impossible to conclude on the role of sialic acid in the M2 phenotype. Furthermore, Siglec-9 KO in monocytes reduced fibroblast-mediated TAM differentiation in co-culture with PS-1 cells, but not in co-culture with M1 or T1 CAFs.

Therefore in the discussion the paragraph is an overinterpretation since this the data show that only in PSC presence the M2 phenotype of TAMs was dependent on sialic acids. " Removing CAF sialylation by sialyltransferase inhibitor treatment or through genetic KO of key regulators such as ST3GAL4 and CMAS that alter the sialylation pathway, resulted in reduced differentiation of monocytes to M2-like TAMs. Our work thus identifies a novel CAF-immune crosstalk dependent on glycosylation of CAFs and the interaction with suppressive receptors on immune cells".

Authors should consider the differences between PSC and CAFs and the involvement of those cells in early stages (ADM acinar to ductal metaplasia) than in late stages (PDAC).

Furthermore, the following conclusion is an overstatement. "Based on marker expression and cytokine profiles, there was no logical explanation for this distinction between PS-1, M1 and T1 CAFs". The activation status of PSC compared to CAF should be taken into account (gel contraction, GFAP expression, T cell inhibition etc..).

We thank the reviewer for their valuable comment. We have now further characterized the different cell lines used in this manuscript by performing glycomics analysis, flow cytometry, western blot and gel contraction assay, which revealed differences between them.

The analysis of glycosylation profiles revealed distinct Siglec-9 ligands between CAF cell lines (Figure 3A-B), which can explain the observed differences in TAM differentiation. Consequently, we included a more nuanced discussion on the results and the complexity of glycosylation-immune interactions [line 508-509, 520-521, 532-535, 601-611, 649]. Additionally, transcriptional data analysis of glycosylation pathways demonstrated enriched N-glycosylation in fibroblast compared to tumor cells (Figure 3C). This underscores the relevance of N-glycosylation in fibroblast, and we were therefore particularly interested in the role of the N-glycan Siglec-9 ligands on PS-1 cells in monocyte-to-macrophage differentiation.

Moreover, differences in the carriers of specific ligands can have an effect in the interaction with Siglec-9 in PS-1 cells compared to M1/T1 CAFs. Supporting this idea, we observed differential expression of several highly glycosylated proteins between CAF cell lines (PDPN, CD106, CD146, supplementary Figure 2F). We consider that the PS-1 may represent the best model to study sialylation in CAFs with an activated phenotype, given the relevance of N-linked glycans in fibroblasts (Figure 3C), coupled with N-linked glycan being the major Siglec-9 ligands in the PS-1 CAF, but not in M1/T1 CAFs (Figure 3B). For clarity of the manuscript, we emphasize the role of sialylation in TAM differentiation in context of PS-1 CAFs, and moved data with other CAF cell lines to supplementary Figure 5.

The remark on the distinction between PSC and CAF roles in early vs late stages is insightful. As mentioned earlier, we have performed additional experiments highlighting that the PS-1 cells are activated myofibroblasts, representative as a model for CAFs.

7. The study lack in vivo relevance of the observed phenotype.

We included the importance of conducting comprehensive in vivo studies for future research in the discussion of the manuscript [line 621-624]. While recognizing the significance of in vivo studies, such experiments were beyond the scope of our current study, which aimed to establish a foundational understanding of CAF sialylation and its interactions with Siglec receptors and myeloid cells. Notably, it's important to acknowledge that Siglec receptors are not conserved between mice and humans, potentially limiting the direct translation of these findings from human to murine in vivo studies.

References

1. van Houtum EJH, Bull C, Cornelissen LAM, Adema GJ. Siglec Signaling in the Tumor Microenvironment. *Front Immunol.* 2021;12:790317.
2. Cha JH, Chan LC, Li CW, Hsu JL, Hung MC. Mechanisms Controlling PD-L1 Expression in Cancer. *Mol Cell.* 2019;76(3):359-70.
3. Beatson R, Graham R, Grundland Freile F, Cozzetto D, Kannambath S, Pfeifer E, et al. Cancer-associated hypersialylated MUC1 drives the differentiation of human monocytes into macrophages with a pathogenic phenotype. *Commun Biol.* 2020;3(1):644.
4. Beatson R, Maurstad G, Picco G, Arulappu A, Coleman J, Wandell HH, et al. The Breast Cancer-Associated Glycoforms of MUC1, MUC1-Tn and sialyl-Tn, Are Expressed in COSMC Wild-Type Cells and Bind the C-Type Lectin MGL. *PLoS One.* 2015;10(5):e0125994.
5. Rodriguez E, Boelaars K, Brown K, Eveline Li RJ, Kruijssen L, Bruijns SCM, et al. Sialic acids in pancreatic cancer cells drive tumour-associated macrophage differentiation via the Siglec receptors Siglec-7 and Siglec-9. *Nat Commun.* 2021;12(1):1270.
6. Verberk SGS, de Goede KE, Gorki FS, van Dierendonck X, Arguello RJ, Van den Bossche J. An integrated toolbox to profile macrophage immunometabolism. *Cell Rep Methods.* 2022;2(4):100192.
7. Beatson R, Tajadura-Ortega V, Achkova D, Picco G, Tsourouktsoglou TD, Klausung S, et al. The mucin MUC1 modulates the tumor immunological microenvironment through engagement of the lectin Siglec-9. *Nat Immunol.* 2016;17(11):1273-81.
8. Rodriguez E, Boelaars K, Brown K, Madunic K, van Ee T, Dijk F, et al. Analysis of the glyco-code in pancreatic ductal adenocarcinoma identifies glycan-mediated immune regulatory circuits. *Commun Biol.* 2022;5(1):41.
9. Wang JHS, Jiang N, Jain A, Lim J. Development of Effective Siglec-9 Antibodies Against Cancer. *Curr Oncol Rep.* 2023;25(1):41-9.
10. Ohlund D, Handly-Santana A, Biffi G, Elyada E, Almeida AS, Ponz-Sarvise M, et al. Distinct populations of inflammatory fibroblasts and myofibroblasts in pancreatic cancer. *J Exp Med.* 2017;214(3):579-96.

REVIEWERS' COMMENTS:

Reviewer #1 (Remarks to the Author):

This reviewer wants to thank the authors for their thorough revisions and additional data provided and considers the manuscript ready for publication. Congratulations.

Reviewer #2 (Remarks to the Author):

All my questions have been addressed adequately by the authors.

Reviewer #3 (Remarks to the Author):

The authors respond to the addressed questions in the revised manuscript. The statistical analysis is appropriate.